# Supply chains create global benefits from improved vaccine accessibility

Daoping Wang [1,2], Ottar N. Bjørnstad [3], Tianyang Lei [4], Yida Sun [4], Jingwen Huo[4], Qi Hao [4], Zhao Zeng [5], Shupeng Zhu [6], Stéphane Hallegatte [7], Ruiyun Li[8], Dabo Guan [4,9] ✉ & Nils C. Stenseth [8,10] ✉

Ensuring a more equitable distribution of vaccines worldwide is an effective strategy to control global pandemics and support economic recovery. We analyze the socioeconomic effects - defined as health gains, lockdown-easing effect, and supply-chain rebuilding benefit - of a set of idealized COVID-19 vaccine distribution scenarios. We find that an equitable vaccine distribution across the world would increase global economic benefits by 11.7% ($950 billion per year), compared to a scenario focusing on vaccinating the entire population within vaccine-producing countries first and then distributing vaccines to non-vaccine-producing countries. With limited doses among low-income countries, prioritizing the elderly who are at high risk of dying, together with the key front-line workforce who are at high risk of exposure is projected to be economically beneficial (e.g., 0.9%~3.4% annual GDP in India). Our results reveal how equitable distributions would cascade more protection of vaccines to people and ways to improve vaccine equity and accessibility globally through international collaboration.

The recurrent waves of SARS-CoV-2 variants have kept the pandemic to continue to threaten public health and society across the globe for three years[1-5]. Though vaccination has regionally mitigated the pandemic toll in certain areas, global inequities in vaccine distribution is an important issue which presently weakens the effectiveness of vaccines in lowering transmission globally[6-8]. Although many world organizations are working to promote accessibility of COVID-19 vaccines through some programs, e.g., the Access to COVID-19 Tools (ACT) Accelerator, the current COVID-19 vaccine distribution across countries shows there are still many disincentives for equitable vaccine distribution globally. Vaccine coverage in many low-income countries is still only around 10%. Given the rapid evolution of SARS-CoV-2[9,10], it is clear that nobody wins the race until everyone wins. This motivates

us to consider the importance of collaboration between vaccine-producing and other countries, guiding future vaccine allocations across countries to allow for a faster recovery of health systems and society. The question is: how do different global vaccine-distribution strategies affect countries' benefits, and how can we design mechanisms to remove disincentives for improving accessibility and equity of vaccines globally? To address these issues, a framework which link epidemiological and socioeconomic modeling frameworks is needed to probe the potential gains of global vaccine allocation strategies from the socioeconomic perspective.

The current local shortages of many commodities even in high-income countries is the clearest illustration of the importance of the highly connected global supply chains. This indicates the cascading

[1]Department of Computer Science and Technology, University of Cambridge, Cambridge, UK. [2]The World Economic Forum, Geneva, Switzerland. [3]Center for Infectious Disease Dynamics, Department of Entomology, Pennsylvania State University, State College, PA, USA. [4]Department of Earth System Science, Tsinghua University, Beijing, China. [5]College of Management and Economics, Tianjin University, Tianjin, China. [6]Advanced Power and Energy Program, University of California, Irvine, Irvine, CA, USA. [7]The World Bank, Washington DC, USA. [8]Centre for Ecological and Evolutionary Synthesis, Department of Biosciences, Faculty of Mathematics and Natural Sciences, University of Oslo, Oslo, Norway. [9]The Bartlett School of Sustainable Construction, University College London, London, UK. [10]Centre for Pandemics and One Health Research, Faculty of Medicine, University of Oslo, Oslo, Norway. ✉e-mail: guandabo@tsinghua.edu.cn; n.c.stenseth@mn.uio.no

effect of pandemic intervention strategies across countries. For example, evidence has shown the negative economic impacts of the lockdown intervention to curb virus transmission in one country spread to other countries along supply chains[1,5,11–13]. From the opposite side, vaccination decisions in one country may be beneficial to the economic recovery of other countries, which is often referred to as one type of externality of vaccination[14–17]. The presence of these externalities is a major driver that makes a market-oriented global vaccine distribution a socially non-optimal solution[18,19]. Advancing our understanding of the positive health and economic externalities is the key to maximize the socioeconomic gains of global vaccine rollout[14,16,18,20].

Here, we quantify the socioeconomic benefits of a set of idealized COVID-19 vaccine-distribution scenarios (Fig. 1) by linking

epidemiological[3,21] and socioeconomic[1,22,23] modeling frameworks. Details of our analytical approach are provided in "Methods". In brief, we base our evaluation on three main outcomes: (i) the health gains, i.e., the value of lives saved through vaccination. Leveraging our realistic age-stratified epidemiological (RAS) model[21], we project burden of mortality averted under varying vaccination scenarios as compared to the "no vaccination" scenario. With the estimates, we used the value of statistical life (VSL) to project the health benefit quantified in US dollars. (ii) the lockdown-easing effect: Assuming that the speed of vaccine rollout is equivalent to the easing of the lockdown[1,21], we multiply the lockdown reduction by sectoral value-added to obtain what we call the lockdown-easing effect (see "Methods" for more details). (iii) the supply-chain rebuilding benefit: We propose a global trade model based on the widely used ARIO

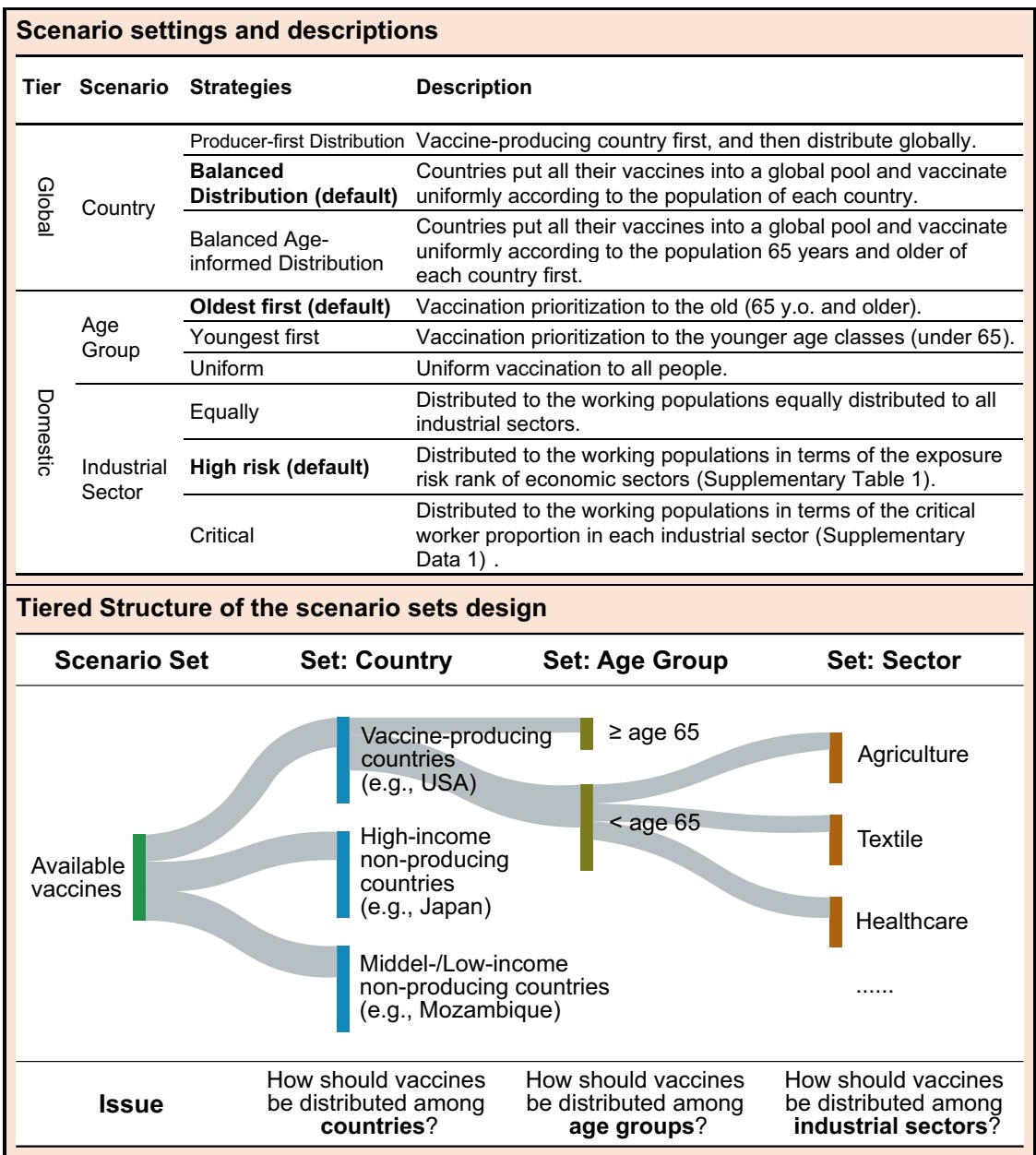

**Fig. 1 | Scenario setting, explanation, and justification.** We designed three sets of scenarios integrated into a tiered structure. Basically, Tier Global set of scenarios addresses the issue of the cooperative attitude of vaccine-producing countries and vaccine-importing countries, while Tier Domestic set of scenarios addresses the issue of how received vaccines (vaccines sent by producing countries) are allocated within each destination country. Among Tier Domestic, sub-scenario A defines the allocation of the received vaccines within destination countries by age group, while sub-scenario S defines the allocation of the received vaccines within destination countries by industrial sectors. One scenario involves a decision at the tiered scenario set, one of which is global and the other 2 domestic.

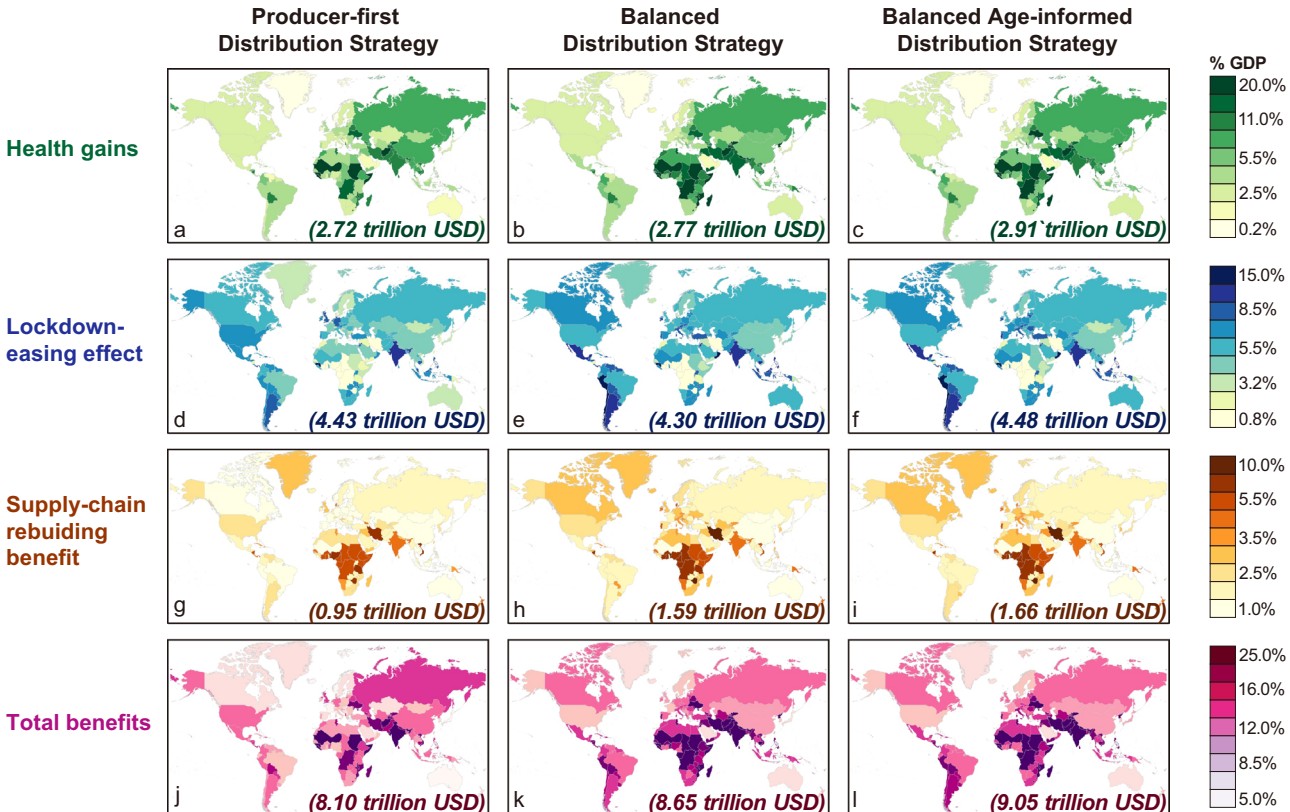

**Fig. 2 | Socioeconomic benefits of vaccination under different global vaccine-distribution strategies.** Each row represents a category of benefit under different scenarios: **a–c** health gains; **d–f** benefits from the alleviation of lockdown; **g–i** benefits from global supply-chain recovery; **j–l** total benefits. Each column represents a global vaccine allocation scenario (see Fig. 1): the left column shows the results under the Producer-first Distribution Strategy; the middle column shows the results under the Balanced Distribution Strategy; and the right column shows the results under the Balanced Age-informed Distribution Strategy. The depth of the color indicates the size of the benefit (expressed as a percentage of the country's GDP). The number in the lower right corner of each map represents the total benefit of the world (expressed as trillion US dollars in 2020). This figure shows the results under the combination of "Oldest First" and "High risk" scenarios, and the results under other scenario combinations are documented in Supplementary Figs. 2–5 and Supplementary Data 2–4.

approach[22,23] to assess the economic losses over 6 years, i.e., 2020–2025, under various vaccination scenarios. We translate the estimates to the total economic benefit brought by the vaccines by quantifying the difference of economic losses with vaccination versus the "no vaccination" scenario. We further subtract the lockdown-easing effect from the total economic benefit to project the supply-chain rebuilding benefit.

We model three sets of idealized scenarios integrated into a tiered structure (see Fig. 1 and Supplementary Fig. 1). Basically, Tier Global set of scenarios addresses the issue of the cooperative attitude of vaccine-producing countries and vaccine-importing countries, while Tier Domestic set of scenarios address the issue of how received vaccines (vaccines sent by producing countries) are allocated within each destination country. We consider different subscenario sets within each scenario set (Fig. 1 and Supplementary Fig. 1). In summary, sub-scenario set Country represents to what extent the vaccine-exporting country is willing to share the vaccine with other countries (specifically, a Producer-first Distribution Strategy vs two Balanced Distribution Strategies, Fig. 1). Subscenario set Age Group defines the allocation of the received vaccines within destination countries by age group (Fig. 1). And subscenario set Sector defines the allocation of the received vaccines within destination countries by industrial sectors (Fig. 1). In the following analysis when comparing the results of one dimension of the scenario sets, "Balanced Distribution Strategy", "Oldest First", and "High Risk" are used as the default scenarios.

## Results

### Supply chains create global economic benefits from vaccine collaboration

Figure 2 summarizes the results of a set of global vaccine-distribution scenarios with a focus on the results under the combination of "Oldest First" and "High risk" scenarios. The results under other scenario combinations are documented in Supplementary Figs. 2–5 and Supplementary Data 2–4. The maps show different types of benefits for 141 modeled regions (see Supplementary Table 2). The panels in the left column (Fig. 2a, d, g, j) show the benefits if the major vaccine-producing countries only distribute vaccines globally after their own population is fully vaccinated (the Producer-first Distribution Strategy); the panels in the middle column (Fig. 2b, e, h, k) show the results when the major vaccine-producing countries share their vaccine with other countries (the Balanced Distribution Strategy; a pure per capita allocation); and panels in the right column (Fig. 2c, f, i, l) show the benefits when the major vaccine-producing countries share their vaccine with other countries with age profile (the Balanced Age-informed Distribution Strategy; an age-adjusted per capita allocation). Three kinds of benefits are shown in the first three rows, namely health gains (Fig. 2a–c), lockdown-easing effect (Fig. 2d–f), and supply-chain rebuilding benefits (Fig. 2g–i). The projected overall benefit is shown in the bottom row of Fig. 2 (Fig. 2j–l).

Altogether Fig. 2 shows that a more equitable distribution of vaccines across the world (i.e., Balanced Distribution Strategies) would bring more societal benefits globally than a vaccine

distribution that is focused on vaccine-producing countries (i.e., the Producer-first Distribution Strategy). If the Producer-first Distribution Strategy is adopted, the total global benefit from vaccination is estimated to be US$8.10 trillion (~9.6% of world GDP) per year. If the Balanced Distribution Strategy is adopted, the total global benefit from vaccination increases to $8.65 trillion (~10.2% world GDP) per year. And if the Balanced Age-informed Distribution Strategy is adopted, the total global benefit from vaccination further increases to $9.05 trillion (~10.7% world GDP) per year. This finding holds under other combinations of domestic scenarios (see Supplementary Figs. 2–5).

A more equitable distribution of vaccines covering a larger number of high-risk populations would not only increase health benefits by protecting more lives and direct domestic production benefits by reducing the need for strict lockdowns but would also facilitate the recovery of the inter-industrial linkages and intra-/inter-regional supply chains (business links between companies). First, compared to the Producer-first Distribution Strategy, the overall health gains have increased by 1.8% under the Balanced Distribution Strategy and 7.0% under the Balanced Age-informed Distribution Strategy (Fig. 2a–c). Under two Balanced Distribution Strategies, the elderly with a high infection-mortality rate and workforce with high exposure risk are covered more, resulting in more lives saved globally (Supplementary Data 5). For example, in Mozambique (one of the least developed economies), the health gains under the Producer-first Distribution Strategy and Balanced Age-informed Distribution Strategy are 9.5% and 13.3% of annual GDP, respectively (Fig. 2a, c). On the other hand, vaccine-producing countries deliver more vaccines to other countries in the two Balanced Distribution Strategies, unsurprisingly leading to a decline in their health gains. For example, in Germany, the health gains under the Balanced Age-informed Distribution Strategy would be reduced by 1.03% of annual GDP compared to the Producer-first Distribution Strategy (Fig. 2a, c). These results of healthy gains show that the marginal health gains of vaccine (i.e., health gains created by each additional unit of vaccine) in countries lacking vaccines are greater than that in countries where vaccines are relatively abundant. Note that, implicit assumptions in the above conclusion are that vaccine supply is the only constraint, while demand is sufficient (e.g., no vaccine hesitancy issue), and distribution processes (e.g., cold chain) are effective.

Second, the overall lockdown-easing effect under the Balanced Distribution Strategy would decrease by 2.9% (Fig. 2d–f), mostly because the economic benefits due to the same degree of lockdown-easing are different between well-developed economies and others representing a lower share of the world GDP. For example, in Germany, benefits from lockdown easing under the Producer-first Distribution Strategy and Balanced Distribution Strategy are 7.4% and 5.1% of annual GDP, respectively (US$82.1 billion decrease; Fig. 2d, e), whereas in Peru, the benefits of the same level of lockdown easing under the Producer-first Distribution Strategy and Balanced Distribution Strategy are 6.9% and 12.3% of annual GDP, respectively (US$10.9 billion increase; Fig. 2d, e). While, the overall lockdown-easing effect under the Balanced Age-informed Distribution Strategy would increase by 1.1% compared to the Producer-first Distribution Strategy (Fig. 2d–f). This is mainly because the balanced allocation strategy has been adjusted according to the age profile of each country, resulting in countries with more elderly people getting more vaccines per capita. These countries with more elderly people also tend to be well-developed economies representing a higher share of the world GDP. Therefore, if only in terms of maximizing direct economic benefits (lockdown-easing effect), the priority of vaccination should be based on GDP per capita. This is very straightforward and seems to reflect the current vaccine-distribution situation. But when we take indirect economic effects (supply-chain rebuilding benefit) into consideration, the results will be different.

Finally, compared to the Producer-first Distribution Strategy, the overall supply-chain rebuilding benefit has increased by 67.4% under the Balanced Distribution Strategy and 74.7% under the Balanced Age-informed Distribution Strategy (Fig. 2g–i). A better recovery within each country under two Balanced Distribution Strategies is crucial to the recovery of the global supply chains. For example, Portugal has a high degree of trade openness (the total of imports and exports as a percentage of GDP, 65.6%), meaning that the country has close supply-chain linkages with other countries across the world. By switching from the Producer-first Distribution Strategy to the Balanced Age-informed Distribution Strategy, Portugal would experience the most significant increase in supply-chain rebuilding effect from 0.9 to 5.0% of annual GDP (Fig. 2g, i). Its largest trading partner, Spain, has a lockdown-easing effect of 6.3% under the Balanced Age-informed Distribution Strategy scenario, which is 46.2% higher than under the Producer-first Distribution Strategy. The recovery of Spain would have a positive spillover effect on Portugal.

In addition to the difference between Producer-first and Balanced Distribution Strategies, our modeling of the two Balanced Distribution Strategies, without- and with age adjustment, shows the potential benefit from considering both the population and age structure of the countries when allocating vaccines internationally. Until now, COVAX has not taken age or disease prevalence into account in country allocation despite strongly urging age-based allocation within countries.

### Benefit-sharing mechanisms that facilitate vaccine cooperation

While our scenarios are idealized cases, the current vaccine-distribution mode is closer to the Producer-first Distribution one[6,7]. Why the equitable vaccine distribution, which promotes global economic benefits, has not been achieved? Answering this question is critical for the response to the ongoing COVID-19 pandemics and, indeed, for future pandemics. Based on our quantification of the vaccine externalities above, we explore opportunities for Pareto improvement (i.e., a reallocation of vaccines and benefits, which can make at least one group better off without making any of them worse off) in the game of global distribution of vaccines. In order to simplify the analysis, we group all countries into three categories: (i) the major vaccine-producing countries; (ii) the non-vaccine-producing countries with high-income levels (>US$4046 per year; World Bank high-income and upper-middle-income countries); and (iii) the non-vaccine-producing countries with low-income levels (<US$4046 per year; World Bank low-middle-income and low-income countries).

Figure 3 shows why equitable vaccine distribution has not been achieved and how a benefit-sharing mechanism may facilitate vaccine cooperation. Flows of money and vaccines without any benefit-sharing mechanism are shown in Fig. 3a, and the corresponding payoff matrix is shown in Fig. 3b. In this case, vaccine-producing countries will choose the Producer-first Distribution Strategy. Because the vaccination-related benefits of the vaccine-producing countries is US$5.31 trillion when they choose the Producer-first Distribution Strategy, higher than the benefits when they choose the Balanced Distribution Strategy (US$4.58 trillion). Naturally, the other two groups of countries were forced to accept the Producer-first Distribution Strategy in this case, although their vaccine-related benefits would increase by 44.8% (high-income nonproducing countries) and 52.7% (low-income nonproducing countries) under the Balanced Distribution Strategy. This dilemma reproduces the current unequal situation, i.e., vaccine-producing countries prefer to give priority to vaccinating their residents, high-income nonproducing countries buy large amounts of vaccines for their domestic use, while middle- and low-income non-producing countries can only obtain very few vaccines due to their insufficient consumption capacity.

Based on the results shown in Fig. 3a, b, it appears clearly why the current global vaccine distribution tends to be the Producer-first Distribution Strategy rather than the Balanced Distribution Strategy, even

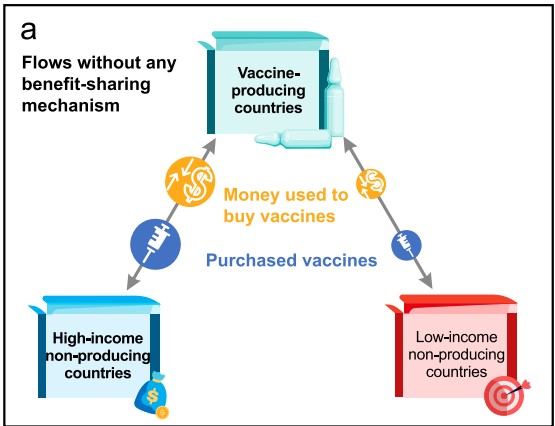

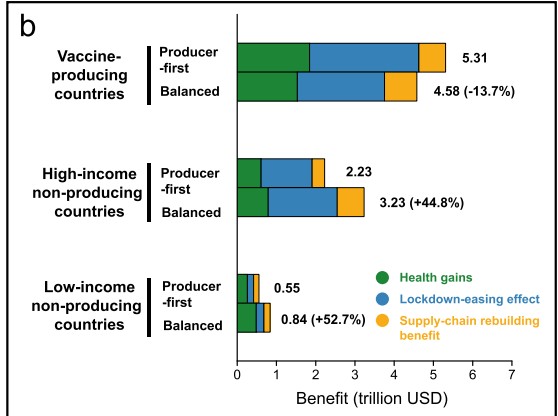

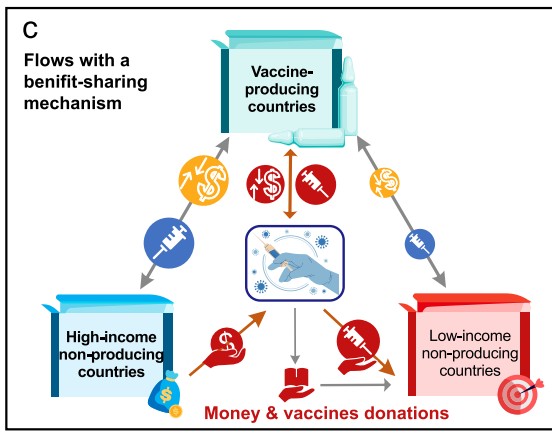

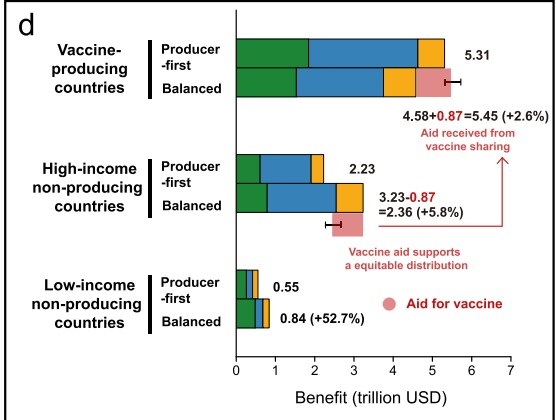

**Fig. 3 | Vaccination-related benefits of three groups under different global vaccine-distribution strategies and the potential incentives (i.e., multilateral benefit-sharing mechanism) to promote a more equitable distribution. a** Shows flows of money and vaccines without any benefit-sharing mechanism; **b** shows the benefits of the three groups of countries without any benefit-sharing mechanism; **c** shows flows of money and vaccines with a benefit-sharing mechanism; **d** shows the new situation (payoffs of the three groups of countries under different distribution strategies) with a benefit-sharing mechanism. The yellow money symbol in (**a**, **c**) indicates money used to buy vaccines and the blue vaccine symbol in (**a**, **c**) indicates vaccines purchased. The red money symbol with a hand below in (**c**) indicates money donations, and red vaccine symbol with a hand below in (**c**) indicates vaccine donations. The "Producer-first" and "Balanced" in (**b**, **d**) represent the "Producer-first Distribution Strategy" scenario and "Balanced Distribution Strategy" scenario. The number on the horizontal bars in (**b**, **d**) indicates vaccination-related benefits (expressed in trillion US dollars). The light red horizontal bars in (**d**) represent the aids from high-income countries to promote a more equitable distribution of vaccines around the world. The "Oldest First" and "High risk" scenario is the default scenario in this comparison.

if only economic benefits are considered (political pressure faced by governments of vaccine-producing countries to prioritize their population before exporting, and even the rise of the so-called "vaccine nationalism[24]" during the COVID-19 pandemic can also be one of the major reasons for the current unequal distribution situation). That is, without any benefit-sharing mechanism, vaccine-producing countries are more willing to choose the Producer-first Distribution Strategy that is most beneficial to themselves, while other countries have no option but to accept an unequal distribution of vaccines.

Figure 3c, d shows how a multilateral benefit-sharing mechanism may incentivize vaccine-producing countries to share vaccines early and promote global vaccine distribution toward a "win-win" equilibrium. Figure 3c shows a multilateral benefit-sharing mechanism, through which, high-income nonproducing countries can donate vaccine aid to the platform to seek a globally equitable distribution of vaccines, vaccine-producing countries deliver vaccines to the platform and obtain corresponding financial returns, and middle- and low-income countries actively cooperate with the platform in completing vaccine delivery and capacity building. Such a benefit-sharing mechanism can be implemented on a global platform (e.g., COVAX).

Figure 3d shows the potential of the multilateral benefit-sharing mechanism shown in Fig. 3c. The benefits of the three groups of countries will be improved simultaneously with the benefit-sharing mechanism when the donation required is within a certain amount.

High-income nonproducing countries can share part of the additional benefits gained as a result of the Balanced Distribution Strategy (e.g., US\$0.87 trillion) with vaccine-producing countries in order to motivate vaccine-producing countries to choose the Balanced Distribution Strategy. If the extra cost is less than US\$1.00 trillion, high-income nonproducing countries are willing to do so because their benefit will be still greater than the benefit in the Producer-first Distribution Strategy after they have paid the cost. Meanwhile, if the transfer is greater than US\$0.73 trillion, vaccine-producing countries will be willing to choose the Balanced Distribution Strategy because their benefit will exceed the benefit of choosing a Producer-first Distribution Strategy (US\$5.31 trillion). And undoubtedly, middle- and low-income countries are willing to receive vaccines or build local production capacity as these are beneficial to them. The basis for this mechanism to become economically rational is the positive externality of vaccination created by global supply chains. When the donation required is within a certain amount, therefore, the three parties are willing to implement this mechanism. Note that the benefit-sharing mechanism shown in Fig. 3c, d only provides a potential economically rational way of international cooperation on the basis of the modeling of the externality of vaccination. It does not mean that the current vaccine distribution is not an economically rational equilibrium, while it highlights that the current situation has room for Pareto improvements through cooperation.

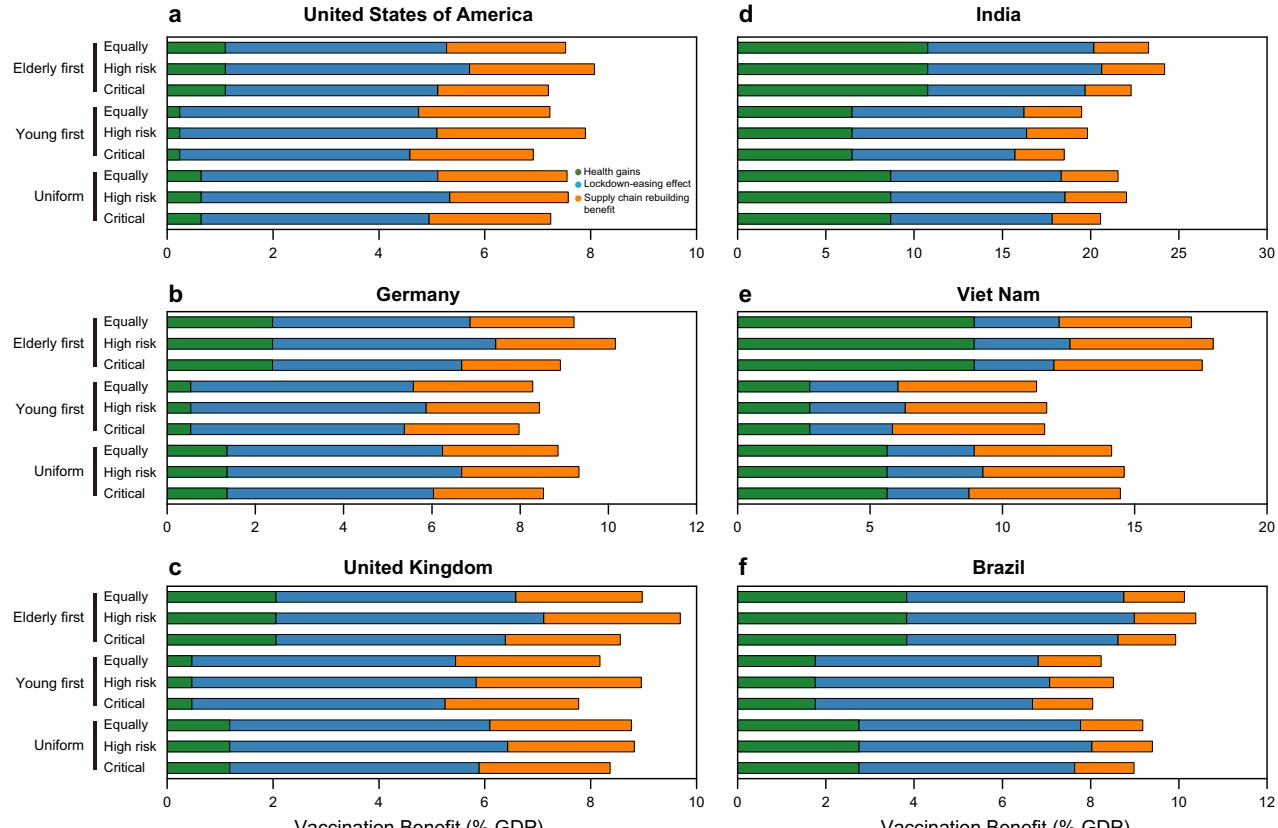

**Fig. 4 | Vaccination-related benefits under different domestic vaccine-distribution strategies for selected countries.** Three types of benefits (health gains, lockdown-easing effect and supply-chain rebuilding benefit) under nine scenarios with different combinations of vaccination priority groups by age and priority of workforce in different industrial sectors (see Fig. 1 and Supplementary Fig. 1). The panels in the left column show the results for three well-developed major vaccine-producing countries (i.e., **a** the United States of America, **b** Germany, and **c** the United Kingdom). The panels in the right column (Fig. 4d–f) show the results for three emerging economies (i.e., **d** India, **e** Vietnam, and **f** Brazil). Colors indicate the category of benefit: the health gains (green), the lockdown-easing effect (blue), and the supply-chain rebuilding benefit (orange). The length of each bar indicates the amount of benefit per year (expressed as a percentage of the country's annual GDP). The "Balanced Distribution Strategy" scenario is the default scenario in this comparison (see Supplementary Data 2–4 for the results of other scenario combinations for all countries).

Our quantification of benefits shows that only when the incentive reaches a certain level can all groups achieve the "win-win" situation. The current proposal made by G7 countries to provide US\$10 billion to COVAX[25] is, however, insufficient to motivate vaccine-producing countries to largely distribute the vaccines to mid- and low-income countries. To ease off the large divide of vaccine distribution, global governance is needed. High-income countries would need to provide necessary capacity building to key personnel in establishing production facilities in mid- and low-income countries, where the local government would need to provide necessary space and tax waiving mechanisms for fast and scale productions in order to minimize the cost of vaccinations (including the manufacturing, transportation, and logistics, and implementing). All these actions that can increase global vaccine production capacity and reduce distribution costs are part or complementary of the multilateral benefit-sharing mechanism shown in Fig. 3c, d.

It is worth mentioning that the manufacturer surplus of vaccine-producing countries is not included in the above discussion. Detailed costs for vaccine producers are difficult to obtain. Moreover, some vaccine-producing companies have pledged to provide their doses on a not-for-profit basis until the pandemic ends. And also, the annual manufacturer surplus is too small (10 billion orders of magnitude) as compared to the health gains and economic benefits from vaccination and supply-chain resumption (1 trillion orders of magnitude). If this is considered, the benefit-sharing mechanism proposed in the present study will be easier to achieve, as vaccine-producing countries will

need less compensation. Hence, when taking the manufacturer surplus into account, all conclusions in the present study will still hold.

## Priority allocation strategies that promote a cascade of vaccine protection in supply chains

Figure 4 depicts the benefits for six representative countries under 9 scenarios with different combinations of vaccination priority groups by age (set of scenario A) and priority of workforce in different industrial sectors (set of scenario S; Fig. 4 shows results under "Balanced Distribution Strategy" scenario for six countries, see Supplementary Data 2–4 for the results of other scenario combinations for all countries; see Supplementary Table 3 for full sector list and Supplementary Table 4 for sector aggregation scheme). The panels in the left column (Fig. 4a–c) show the results for three well-developed major vaccine-producing countries (i.e., the United States of America, Germany, and the United Kingdom). The panels in the right column (Fig. 4d–f) show the results for three emerging economies (i.e., India, Vietnam, and Brazil).

Figure 4 shows that economic benefits are largest when the priority of domestic vaccine distribution is given to the elderly segment, i.e., 65-years-old and above, of the population, followed by workforce with high exposure risk, such as workers in transportation, accommodation, and catering industrial sectors. The figure shows that giving priority to the elderly generally provides higher economic benefits, even though the difference is not as large as could have been expected. For example, the total benefit from vaccination in the USA is

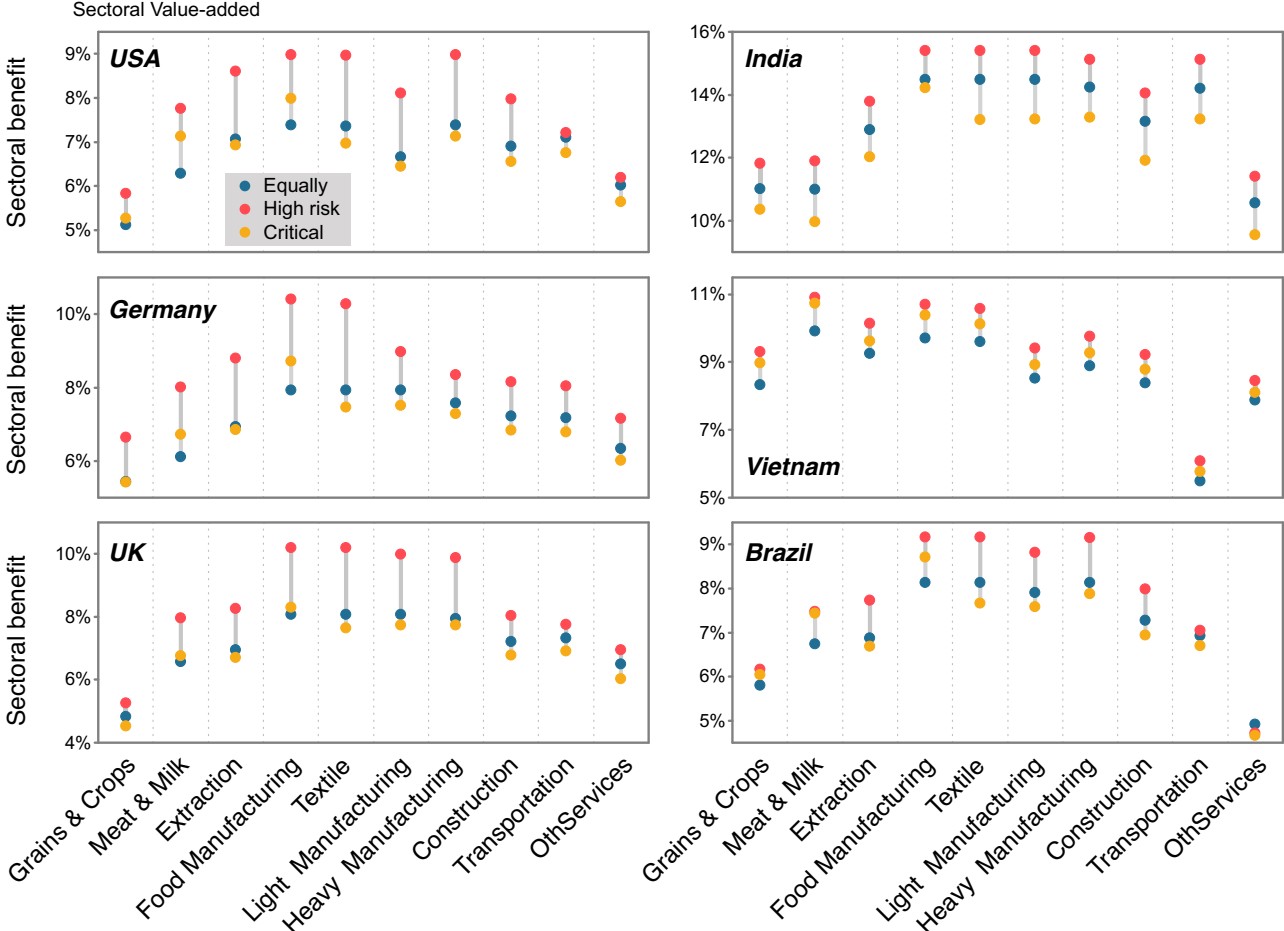

**Fig. 5 | Sectoral benefits under three scenarios with different vaccination priorities of industrial sectors.** The x-axis represents ten industrial sectors (see Supplementary Table 4 for sectoral information), and the y-axis represents the total economic benefit of each industrial sector (expressed as a percentage of the value-added of the corresponding industrial sector). Color represents different vaccination scenarios: blue, any available dose will be equally allocated from the outset to the working populations across all economic sectors (Equally); red, any available dose will be first given to the working populations of specific sectors as ranked in terms of the exposure risk (High risk); orange, any available dose will be first given to the working populations in each economic sector based on the proportion of critical workers (Critical, total labor requirements to meet demand for basic necessities, see Supplementary Data 1). The "Balanced Distribution" and "Oldest First" scenarios are the default scenario in this comparison.

about 7.5% (7.2–8.1%) of annual GDP when the elderly group is prioritized, whereas the total benefits decrease to about 7.2% (6.9–7.9%) of annual GDP when young people are prioritized. This reduction is mainly explained by the difference in health gains from vaccination. The health benefit from vaccination of the USA population is 1.1% of annual GDP when the vaccine is given first to the elderly, about four times higher than the health benefit if priority is given to young people (0.2% of USA annual GDP). The results of other countries also support this conclusion (Fig. 4b–f and Supplementary Data 2–4). The infection-mortality rate among the elderly is relatively high. Prioritizing the elderly can thus save more lives (see Supplementary Data 5) and result in higher health gains.

Once the elderly is fully vaccinated, moving to vaccinate the workforce in sectors with high exposure risk would bring higher economic benefits than an equally distribution across sectors ("Equally" scenario), through both lockdown-easing effect and supply-chain rebuilding benefit (Fig. 4). For the United States, Germany, Vietnam, and Brazil, this pattern is more pronounced. For example, the lockdown-easing effect from vaccination in the USA is 4.6% of annual GDP when workforce with high exposure risk is prioritized (note that Balanced Distribution Strategy and Oldest First are the default when we discuss results of sectoral distribution), 10.3% higher than the lockdown-easing effect if vaccinating the whole workforce equally

(4.2% of annual GDP). Meanwhile, the supply-chain rebuilding benefit from vaccination in the USA is 2.4% of annual GDP when the vaccine is given first to workforce with high exposure risk, 9.1% higher than the lockdown-easing effect if vaccinating the whole workforce equally (2.2% of annual GDP). The results of other countries also support this conclusion (Fig. 4b–f and Supplementary Data 2–4). Prioritizing vaccination of workers with high exposure risk can greatly reduce the need of strict lockdown, which is conducive to the recovery of the local economy, and further the global supply chains.

Figure 5 represents the sectoral benefits under different domestic vaccine-distribution strategies, shows that prioritizing workforce with high exposure risk (Supplementary Table 1), would maximize economic recovery in those sectors but also create strong positive spillover effects to other production sectors. Generally, the entire economy would obtain the largest benefits when workers in production sectors at high risk of exposure are prioritized (scenario S−High risk; red dots in Fig. 5)—this being due to the inter-sector spillovers[26–28]. Prioritizing high-risk groups (red dots in Fig. 5) contributed 0.1–10.9% extra spillover benefits as compared with an equal distribution strategy across industrial sectors (blue dots in Fig. 5; see Supplementary Data 6). This highlights the importance of considering externalities when designing domestic vaccine allocation strategies.

The tighter the association among domestic sectors, the larger the spillover benefits from the recovery of one sector to all other economic sectors. In well-developed economies, i.e., the USA, Germany, and the UK, the association among domestic sectors is relatively high. Therefore, giving priority to workforce with high exposure risk not only help the rapid recovery of these sectors, but the recovery effect will quickly cascade to other sectors. For example, in the USA and Germany, the benefits of "Grains and Crops" sector would increase by 15.2% and 23.6%, respectively, under "High risk" scenario compared to 'Equally' scenario. This is because machinery is a key input in modern agriculture in well-developed countries[29,30]. The recovery of light and heavy manufacturing industries is conducive to the recovery of the agricultural sector, which may otherwise suffer, e.g., from missing parts for equipment. Given that in low-income countries, the association among sectors along the domestic supply chains is weaker than in developed countries, the spillover benefits would be lower than in high-income countries. For example, Viet Nam and in Brazil would see the spillover benefit to the agriculture sector increasing by only 11.9% and 6.6%, respectively, under "High risk" scenario compared to "Equally" scenario.

We also analyzed the benefits under the "Critical" scenario (orange dots in Fig. 5) in which any available dose will be firstly given to the working populations in terms of the critical worker proportion in each industrial sector (see Supplementary Data 1). For example, compared to the recreation sector, workers in the food manufacturing sector will be given priority. Figure 5 shows that the benefits of the "Critical" scenario (orange dots) are generally the lowest among the three vaccination strategies, which indicates that there is a trade-off between guaranteeing food/daily necessities and overall economic recovery. Therefore, in the most urgent situation, we should give priority to workforce in the critical sector. But after the food and necessities are met, we should give priority to workforce in high-risk sectors to optimize the economic recovery.

## Discussion

By quantifying the economic benefits, including health gains, lockdown-easing effect, and supply-chain rebuilding benefit, of each player (country/region) under a set of vaccine-distribution scenarios designed in the study, we provide new information to understand the game of global vaccine distribution and facilitate global vaccine cooperation. Our results demonstrate the potentially significant differences in the socioeconomic benefits brought by different global and domestic vaccines distribution modes. The protection of vaccines has strong spillover effects in supply chains, and a more equitable distribution helps supply chains create more global benefits.

Our analysis also reveals why an equitable vaccine distribution, which promotes global economic benefits, has not been achieved and shows how a multilateral benefit-sharing mechanism may facilitate global vaccine cooperation. The "equitable distribution", in this study, is not a simple appeal, but a solution that makes economic sense. This is not about donors sacrificing themselves for global economic benefit. The benefit-sharing mechanism proposed in this study allows all players to benefit from "equitable distribution" simultaneously. This is one of the most important lessons we should learn from the COVID-19 pandemic. Moreover, the method developed in this study provides a model for the analysis of such complex problems. This analysis framework allows us to analyze vaccine-distribution strategies in different infectious and socioeconomic contexts rather than just for COVID-19.

It is worth pointing out, although the difficulty of global collaboration itself is a reason why there is not enough benefit sharing to motivate an equitable distribution currently, we believe that a deeper reason may be lacking complete information about the payoff matrix of the global vaccine-distribution game. The benefits of health (health gains) are straightforward and easy to be taken into consideration. The

benefits of the economic recovery (lockdown-easing effect and supply-chain rebuilding benefit), however, are often not well quantified and considered. This situation will lead to bias in the player's decision-making. A key significance of the quantitative analysis presented in this study is that it provides countries with a comprehensive understanding of their potential payoffs in the global vaccine-distribution game, which is a prerequisite for players to make the right decision.

Our comprehensive quantification may also help in easing political pressure faced by governments of vaccine-producing countries to prioritize their population before exporting, promoting an equitable distribution from the political aspect. If only health benefits are considered, the public will see the sharing of vaccines with other countries as a pure sacrifice when domestic vaccine demand has not been fulfilled. But it will not the case when the supply-chain benefits quantified in this study are considered, i.e., sharing vaccines benefits both the recipient and the sharer.

Our study has some limitations. We do not build a feedback mechanism between the epidemiological and the economic model, i.e., the interaction between the intensity of economic activity and the spread of the virus. We acknowledge that a feedback mechanism is theoretically feasible. The current practical knowledge in this area, however, is still very limited, which means that the introduction of a feedback mechanism will bring about very large uncertainties. Our model is also limited by taking no consideration of technological changes and adjustment of behaviors and by assuming that production and consumption patterns remain the same as pre-crisis. Our model has a focus on short-term scenarios, and therefore the above two assumptions are rather unlikely to have a significant impact on the results[1]. Our model is further constrained by the trade relationship at the sectoral level among countries, and has no ability to capture the complexity of supply-chain networks at the firm level and may therefore underestimate the benefits[31]. In addition, this study only focuses on economic benefits. We acknowledge that maximizing the aggregate benefit is not the only criteria that need to be accounted for, but also fairness and feasibility. But these are beyond the scope of this study. In addition, exploring where/which regional locations will be adding more production capacity contribute the most to the objective function of the global allocation is also an important and interest policy question but not include in the present study. This is an area worth exploring in the future.

A more general significance of this study is that it provides a new perspective to understand the relationship between efficiency and equity on the way to sustainable development. The idea that there is a trade-off between equity and efficiency sometimes may be ingrained in our subconscious. We, therefore, tend to always think about how to balance efficiency and equity in making decisions. Our analysis, using COVID-19 and the global vaccine-distribution game as an example, shows that there are some cases in which equity is more efficient. Comprehensive quantification and carefully designed mechanisms, as done in this study, would be of great help to the synergy of equity and efficiency on the way to achieving the Sustainable Development Goals.

In preparing for future pandemics, a multilateral benefit-sharing instrument should be developed so as to remove some of the disincentives for early equitable vaccines distribution globally. Such an instrument would provide enormous global health and economic benefits in a sustainable manner.

## Methods

### Vaccine-distribution scenario sets

To evaluate the economic benefits of allocating the vaccines across the globe, we propose three scenario sets which are designated in a tiered structure (see Fig. 1 and Supplementary Fig. 1). Basically, Tier Global scenario sets address the issue of the cooperative attitude of vaccine-exporting countries and importing countries, while Tier Domestic scenario sets address the issue of allocating the received vaccines

within the destination countries. We treat each scenario set as an individual parameter in the model, such that we will have three parameters (i.e., Country, Age Group, Sector). We vary the value of each parameter by considering different sub-scenarios within each scenario set. In summary, parameter Country indicates whether the vaccine-exporting country is more willing to share the vaccine with other countries. Parameter Age Group define the allocation of the received vaccines within destination countries according to the age. And parameter Sector define the allocation of the received vaccines within workforce in destination countries according to the feature of industrial sector.

It is worth noting that countries/regions in this article do not only refer to governments. When we say a country, we mean to abstract this country into a representative agent. For example, vaccines are generally produced by private firms. But for the convenience of discussion, we abstract all private firms combined with all other economic participant in vaccine production into an agent.

**Global vaccine distribution (tier global).** Scenario set C (Country): what extent the vaccine-exporting country is willing to share the vaccine with other countries? The acronym $C$ is "Country".
- Producer-first Distribution Strategy. Countries producing vaccines use production to fully vaccinate their own population first, and then distribute globally.
- Balanced Distribution Strategy. Countries put all their vaccines into a global pool and vaccinate uniformly.
- Balanced Age-informed Distribution Strategy. Countries put all their vaccines into a global pool and vaccinate uniformly according to the population 65 years and older of each country first, if sufficient, then vaccinate uniformly according to the population under 65-years-old (y.o., hereafter) of each country.

**Domestic vaccine distribution (tier domestic).** Scenario set A (Age group): prioritized age groups in the destination countries. Old first or Youngest First. The acronym $A$ is "age".
- Oldest First: Vaccination prioritization to the old (over 65 y.o.). We assume a mass vaccination for the 65+. If sufficient, the doses will then be distributed to people between 20 to 65 y.o. This follows the schedule in most countries.
- Youngest First: Vaccination prioritization to the younger age classes (20–65 y.o.). We assume a mass vaccination for people between 20 to 65 y.o. If sufficient, the doses will then be distributed to the old over 65 y.o. This follows the schedule in China.
- Uniform: Mass vaccination to all people over 20 y.o.

Scenario set S (Industrial sector): prioritized socially vulnerable groups in the destination countries. Considering labors in different sectors, critical workers first or mass distribution. The acronym $S$ is "industrial sector".
- High risk: Any available dose will be firstly given to the working populations in terms of the exposure risk rank of economic sectors (see Supplementary Table 1).
- Equally: Any available dose will be firstly given to the working populations equally distributed to all economic sectors.
- Critical: Any available dose will be firstly given to the working populations in terms of the critical worker proportion in each economic sector (see Supplementary Data 1).

The combination of each variation of above three parameters gives a distribution strategy. In this analysis, we will have 21 scenarios (the "Balanced Age-informed Distribution Strategy" implies the "Oldest First"). When comparing the results of a scenario set, "Balanced Distribution", "Oldest First", and "High risk" are used as the default scenario.

## Estimation of vaccine production capacity
We consider seven major vaccine-manufacturing counties, including China, USA, Germany, India, UK, the Netherlands, and Russia[32]. We collected the current vaccine production capacity of these countries, and based on this, we predicted the future vaccine production capacity.
- The overall capacity of all manufacturing countries in the "Approved in use" development stage and in the future (i.e., 2022–2023) is collected from the United Nations International Children's Emergency Fund (UNICEF)[32] (see Supplementary Figs. 6 and 7 and Supplementary Table 5).
- With the data, we project the annual capacity of all manufacturing countries by using the logarithmic function to fit the growth trend of capacity (Supplementary Fig. 6).
- Assuming an invariant relative capacity across countries over time, we further partition the annual capacity to each of the manufacturing country according to their capacity documented on March 3, 2021 (Supplementary Table 5 and Supplementary Fig. 7). Supplementary Figs. 8 and 9 show the average doses of vaccine available per capita per year in each country/region during 2020–2025.

It is important to note that COVID-19 and vaccine-manufacturing capabilities are changing rapidly. The vaccine production capacity pathway used in this study is only the best estimates when the study carried out and an application case of the proposed approach. Considering that COVID-19 and vaccine production capacity is still changing rapidly and the different starting points for future pandemics, this study aims to provide a general methodology for analysis of this kind of problems. Analysis of various vaccine production capacity pathways and future pandemic scenarios could be performed in the future using the methods developed in this study.

## The epidemiological model
**Model structure.** Built upon our age-structured SIR model[3,21], we project the fraction of incidence and mortality over age groups by using chains of differential equations:

$$\frac{dS_i^p}{dt} = a_{i-1}S_{i-1}^p - \lambda_i S_i^p - a_i S_i^p \tag{1}$$

$$\frac{dI_i^p}{dt} = a_{i-1}I_{i-1}^p + \lambda_i S_i^p - \gamma I_i^p - a_i I_i^p \tag{2}$$

$$\frac{dR_i}{dt} = a_{i-1}R_{i-1} + \gamma(I_i^p + I_i^{np}) - \omega R_i - a_i R_i \tag{3}$$

$$\frac{dS_i^{np}}{dt} = \omega R_i + a_{i-1}S_{i-1}^{np} - \lambda_i S_i^{np} - a_i S_i^{np} \tag{4}$$

$$\frac{dI_i^{np}}{dt} = a_{i-1}I_{i-1}^{np} + \lambda_i S_i^{np} - \gamma I_i^{np} - a_i I_i^{np} \tag{5}$$

where $S_i^p, I_i^p$ are the number of susceptible individuals and primary infections in age group $i$; accordingly, $I_i^{np}$ is the number of non-primary infections. The recovered individuals ($R_i$) may lose immunity and return to susceptibility ($S_i^{np}$) after an average duration of immunity of $1/\omega$. In the case that $\lambda_i$ and $\gamma$ are the same for first infection and subsequent infections, we do not need to distinguish between $S_i^{np}$ and $I_i^{np}$. We set the model as in Eq. (1)–(5) in order to keep the generality of the model, which equipped the model with the ability to handle the case that $\lambda_i$ and $\gamma$ are different for first and subsequent infection. The force-of-infection on susceptible in age-class $i$ is designated as

$\lambda_i = \beta \sum_j^n C_{ij}(I_i^p + I_i^{np})/N_i$, where $\beta = R_0\gamma$ is the baseline rate of transmission and $C_{ij}$ is the contact rate between age group $i$ and $j$. $1/\gamma$ to be the average duration of infection which is taken to be 7 days[33]. For simplicity, we assume a uniform 1-year duration rate of aging ($a_i$) across ages, i.e., $a_i = 1$ for all $i$. For $R_0$, we refer to Liu et al.[34] in which identified 12 studies which estimated the basic reproductive number for COVID-19 from China and overseas and shown that the estimates ranged from 1.4 to 6.49. Assuming $R_0 = 3.5$, we parameterize the model with country-specific population pyramid[35] and social mixing[36] over 16 age groups. Details of model parameters is provided in Supplementary Table 6.

For the vaccine intervention case, based on the model shown in Eqs. (1)–(5), we developed the following realistic age-structured multi-compartmental SEIR model that allows for projections of disease burden of SARS-CoV-2 virus with diverse intervention strategies,

$$\frac{dS_i}{dt} = \omega R_i - \lambda_i S_i - qQ_i S_i \tag{6}$$

$$\frac{dE_i}{dt} = \lambda_i S_i - \delta E_i \tag{7}$$

$$\frac{dI_i}{dt} = \delta E_i - \gamma I_i \tag{8}$$

$$\frac{dR_i}{dt} = \gamma I_i + qQ_i S_i - \omega R_i \tag{9}$$

where $S_i$, $E_i$, $I_i$ and $R_i$ are, respectively, the number of susceptible, exposed, infected and recovered individuals in age group $i$. The recovered/vaccinated individuals are assumed to lose immunity and return to susceptibility after an average protected period of $1/\omega$ and subsequently be liable to reinfection. The average incubation period $1/\delta$ in the analysis is taken to be 5.2 days[33]. To appropriately model the *fraction* of individuals vaccinated, we define the *rate* of vaccination ($Q_i$) as $-\log(1 - P_i)/D_i$ where $P_i$ and $D_i$ is the vaccine coverage and duration of vaccination in age group $i$, respectively[37,38]; $q$ is the vaccine efficacy, take the value 50%.

**Model simulation.** Simulation was initialized with 1% infections and 0.1% recovered individuals, i.e., $S^p(0) = 0.989, I^p(0) = 0.01, R = 0.001$, and $S^{np}(0) = I^{np}(0) = 0$. All rate parameters have units "per day". We project the model to predict dynamics of COVID-19 in the next 6 under different vaccine allocation strategies. We do not explicitly model the timing of the vaccination campaign; instead, we assume that vaccines are uniformly distributed over the year. With the simulation we estimate the age-specific fraction of infection and further infer the fraction of deaths by multiplying the age-infections with infection fatality ratio (IFR)[39]. Considering the current spread and variation of COVID-19, we used the current infection and death data to scale up the simulation results. By comparing the results of scenarios with or without vaccines, we can obtain the benefits of different vaccine-distribution strategies.

**Model assumptions.** To appropriately lay out our insights, we make several assumptions. First, we assume the homogeneous susceptibility to infection, clinical fraction and infection vs case-fatality ratio as well as the immunity-dependent infectiousness of reinfection across age classes. Additionally, we assume a 1-year duration of immunity, given the brief immunity from natural infection of seasonal coronavirus[40]. Moreover, we assume a uniform distribution of vaccine rollout over the year. Relaxing the assumptions by explicitly consider age-specific heterogeneities, differing durations of immunity and the timing of

vaccination are easy extensions giving the general nature of our model framework.

**The estimation of value of statistical life (VSL)**
The value of statistical life (VSL) is widely used throughout the world to monetize fatality risks in benefit-cost analyses[41]. The VSL represents the individual's local money-mortality risk trade-off value[42], which is the value of small changes in risk not the value attached to identified lives.

Given that the effect of age on VSL is theoretically indeterminate and that the empirical evidence is mixed, experts and public agencies around the world are split as to whether the VSL should be adjusted in assessing health benefits and costs for people of different ages. Using a unified VSL for all countries is consistent with the desire to value each life equally for international donations of COVAX.

The country-based VSL estimation used in this research is adopted from the COVID-19 global health risks pricing study by Viscusi Table 6 in ref. [43]. The estimation is based on the estimated VSL in the U.S. (11 million in 2019 US dollar). Based on this, we do two VSL estimation. We first use an income elasticity (=1.0) to adjust the VSL to other countries using the fixed effects specification[44]. Supplementary Fig. 10, shows the spatial distribution of estimated VSL for 175 countries used in this approach. And then, to keep in line with the idea that every life is equal, we value each life equally with a global average a uniform global VSL (=11 USD million times average global GDP per capita/US GDP per capita; 2.94 USD million).

**Estimation of required strictness of control measures**
COVID-19 has resulted in varying degrees of social lockdown in countries all around the world[45]. The lockdown strictness by each country is measured by the percentage by which labor availability and transportation capacity are reduced relative to pre-pandemic levels. The Google Community Mobility Reports (COVID-19 Community Mobility Reports), which aim to provide insights into changes in response to policies aimed at combating COVID-19, are used to measure the strictness specifically. The reports chart movement trends over time by geography, across different categories of places such as retail and recreation, groceries and pharmacies, parks, transit stations, workplaces, and residential. We averaged the changes in the five types of visitors as a parameter to measure the extent of a country's lockdown strictness. The data are monthly, starting in February 2020 and the latest up to April 2021. For countries where Google data are not available, we supplement them with data from the nearest country based on geographical location.

The intensity of a country or region's lockdown at any period is determined by three factors: its initial lockdown intensity, the size of its population, and the number of people it currently has protected. Among them, the initial lockdown strength is derived from Google Community Mobility Reports. Population sizes were obtained from the World Bank database. The number of people already protected is calculated from scenarios and our epidemiological model.

Note that, the lockdown in the present study does not distinguish between voluntary self-protection and formal lockdown. It is because we use Google mobility data (COVID-19 Community Mobility Reports) as an indicator of the intensity of the lockdown. This indicator is the combined result of the two effects (voluntary behavior change and formal lockdown). Hence, this study takes the possibility of the spontaneous behavior of economic agents into account naturally.

**The recursive dynamic disaster impact assessment model for estimation of lockdown-easing effects and supply-chain rebuilding benefit**
In addition to the life-saving benefits calculated by infectious disease models, vaccine distribution also generates lockdown-easing effects

and supply-chain rebuilding benefits through the global supply chains. The global economic benefits will be calculated using the recursive dynamic disaster impact assessment model[1,22,23,46].

Our disaster impact assessment model is an extension of the adaptive regional input–output (ARIO) model[22,23], which was widely used in the literature to simulate the propagation of negative shocks throughout the economy[1,5,31,47]. Input–output analysis (IOA) has been proved a very powerful economic analysis tool. It captures the economic linkages between countries and industrial sectors, which makes it well-suited for studying the economic externalities of vaccine distribution. The recursive dynamic disaster impact assessment model is an improved input–output model which can better describe the economic dynamics after the disaster. It has been wildly used in the post-disaster economic dynamic analysis, such as refs. [31,48–50]. Take Inoue and Todo[31] as an example, it uses the model to simulate the economic dynamics after the 2011 Japan earthquake and their results fit the real economic dynamics well. The simulation code and examples can be found in GitHub (https://github.com/DaopingW/economic-impact-model).

Our model improves the ARIO model in two ways. The first improvement is related to the substitutability of products from the same sector sourced from different regions (see description of "Production module" of the model). Second, in our model, clients will choose their suppliers across regions based on their capacity (see description of "Demand module" of the model). These two improvements contribute to a more realistic representation of bottlenecks along global supply chains.

Our disaster impact assessment model mainly includes four modules, i.e., production module, allocation module, demand module and simulation module. The production module is mainly designed to characterize the firm's production activities. The allocation module is mainly used to describe how firms allocate output to their clients, including downstream firms (intermediate demand) and households (final demand). The demand module is mainly used to describe how clients place orders to their suppliers. And the simulation module is mainly designed for executing the whole simulation procedure.

**Production module.** The production module is used to characterize production processes. Firms rent capital and employ labor to process natural resources and intermediate inputs produced by other firms into a specific product (see Supplementary Fig. 11). The production process for firm $i$ can be expressed as follows,

$$x_i = f\left(\text{for all } p, z_i^p; \text{va}_i\right) \tag{10}$$

where $x_i$ denotes the output of the firm, in monetary value; $p$ denotes type of intermediate products; $z_i^p$ denotes intermediate products used in production processes; $\text{va}_i$ denotes the primary inputs to production, such as labor ($L$), capital ($K$) and natural resources (NR). $f(\bullet)$ is the production function for firms. There are a wide range of functional forms, such as Leontief[46], Cobb–Douglas (C-D) and Constant Elasticity of Substitution (CES) production function[51]. Different functional forms reflect the possibility for firms to substitute an input for another. Considering that epidemics often cause large-scale economic fluctuations in the short term, during which economic agents do not have enough time to adjust other inputs to substitute temporary shortages, we use Leontief production function which does not allow substitution between inputs.

$$x_i = \min\left(\text{for all } p, \frac{z_i^p}{a_i^p}; \frac{\text{va}_i}{b_i}\right) \tag{11}$$

where $a_i^p$ and $b_i$ are the input coefficients calculated as

$$a_i^p = \frac{\bar{z}_i^p}{\bar{x}_i} \tag{12}$$

and

$$b_i = \frac{\overline{\text{va}}_i}{\bar{x}_i} \tag{13}$$

where the horizontal bar indicates the value of that variable in the equilibrium state. This production function allows products of the same industrial sector in different regions to be substituted for each other. In other words, products from same industrial sector of different regions will go into the same inventory.

In an equilibrium state, producers use intermediate products and primary inputs to produce goods and services to satisfy demand from their clients. After a disaster, output will decline. From a production perspective, there are mainly the following constraints:

**Labor supply constraints.** Labor constraints after a disaster may impose severe knock-on effects on the rest of the economy[22,52]. This makes labor constraints a key factor to consider in disaster impact analysis. For example, in the case of a pandemic, these constraints can arise from employees' inability to work as a result of illness or death, or from the inability to go to work and the requirement to work at home (if possible). In this model, the proportion of surviving productive capacity from the constrained labor productive capacity ($x_i^L$) after a shock is defined as:

$$x_i^L(t) = \left(1 - \gamma_i^L(t)\right) * \bar{x}_i \tag{14}$$

Where $\gamma_i^L(t)$ is the proportion of labor that is unavailable at each time step $t$ ($t$ is 2020 to 2025 in this study) during containment. $\left(1 - \gamma_i^L(t)\right)$ contains the available proportion of employment at time $t$.

$$\gamma_i^L(t) = \left(\bar{L}_i - L_i(t)\right)/\bar{L}_i \tag{15}$$

We estimate the without-vaccine $\gamma_i^L(t)$ based on data from Google Community Mobility Reports. The reports chart movement trends over time by geography, across different categories of places such as retail and recreation, groceries and pharmacies, parks, transit stations, workplaces, and residential. We averaged the changes in the five types of visitors as a parameter to measure the extent of a country's lockdown strictness. The data are monthly, starting in February 2020 and the latest up to April 2021. For the with-vaccine case, we assume that the constrained labor supply will ease linearly with vaccine coverage. More intuitively, the people protected by the vaccine can end his isolation and he can start to work. Therefore, $\gamma_i^L(t)$ will decrease linearly with vaccine coverage.

The proportion of the available productive capacity of labor is thus a function of the losses from the sectoral labor forces and its pre-disaster employment level. Following the assumption of the fixed proportion of production functions, the productive capacity of labor in each region after a disaster ($x_i^L$) will represent a linear proportion of the available labor capacity at each time step. Take COVID-19 as an example, during an outbreak of an infectious disease, authorities often adopt social distancing and other measures to reduce the risk of infection. This imposes an exogenous negative shock on the economic network.

**Constraints on productive capital.** Similar to labor constraints, the productive capacity of industrial capital in each region during the aftermath of a disaster ($x_i^K$) will be constrained by the surviving capacity of the industrial capital[47,53,54]. The share of damage to each

sector is directly considered as the proportion of the monetized damage to capital assets in relation to the total value of industrial capital for each sector, which is disclosed in the event account vector (EAV) for each region ($\gamma_i^K$), following[54]. This assumption is embodied in the essence of the IO model, which is hard-coded through the Leontief-type production function and its restricted substitution. That is, as capital and labor are considered perfectly complementary as well as the main production factors, and the full employment of those factors in the economy is also assumed, we assume that damage in capital assets is directly related with production level and therefore, value-added level. Then, the remaining productive capacity of the industrial capital at each time step is defined as:

$$x_i^K(t) = \left(1 - \gamma_i^K(t)\right) * \bar{x}_i \tag{16}$$

where, $\bar{K}_i$ is the capital stock of firm $i$ in the pre-disaster situation, and $K_i(t)$ is the surviving capital stock of firm $i$ at time $t$ during the recovery process.

$$\gamma_i^K(t) = \left(\bar{K}_i - K_i(t)\right) / \bar{K}_i \tag{17}$$

**Supply constraints.** Firms will purchase intermediate products from their supplier in each period. Insufficient inventory of a firm's intermediate products will create a bottleneck for production activities. The potential production level that the inventory of the $p^{\text{th}}$ intermediate product can support is

$$x_i^p(t) = \frac{S_i^p(t-1)}{a_i^p} \tag{18}$$

where $S_i^p(t-1)$ refers to the amount of $p^{\text{th}}$ intermediate products held by firm $i$ at the end of time step $t-1$.

Considering all the limitation mentioned above, the maximum supply capacity of firm $i$ can be expressed as

$$x_i^{\text{max}}(t) = \min\left(x_i^L(t); x_i^K(t); \text{for all } p, x_i^p(t)\right) \tag{19}$$

Considering that lockdowns only cause labor supply constraint without destroying capital, $\left(\bar{x}_i - x_i^{\text{max}}(t)\right)$ can represents the loss of output due to direct labor supply constraints.

The actual production of firm $i$, $x_i^a(t)$, depends on both its maximum supply capacity and the total orders the firm received from its clients (see the "Demand module"),

$$x_i^a(t) = \min\left(x_i^{\text{max}}(t), \text{TD}_i(t-1)\right) \tag{20}$$

The inventory held by firm $i$ will be consumed during the production process,

$$S_i^{p,\text{used}}(t) = a_i^p * x_i^a(t) \tag{21}$$

**Allocation module.** The allocation module mainly describes how suppliers allocate products to their clients. When some firms in the economic system suffer a negative shock, their production will be constrained by a shortage to primary inputs such as a shortage of labor supply in the outbreak of COVID-19. In this case, a firm's output will not be able to fill all orders of its clients. A rationing scheme that reflects a mechanism based on which a firm allocates an insufficient amount of products to its clients is needed. For this case study, we applied a proportional rationing scheme according to which a firm allocates its output in proportion to its orders. Under the proportional rationing scheme, the amounts of products of firm $i$ allocated to firm $j$ and household $h$ is as follows (see Supplementary Text and Supplementary

Table 7 for a sensitivity analysis for different allocation modes),

$$\text{FRC}_j^i(t) = \frac{\text{FOD}_i^j(t-1)}{\left(\sum_j \text{FOD}_i^j(t-1) + \sum_h \text{HOD}_i^h(t-1)\right)} * x_i^a(t) \tag{22}$$

$$\text{HRC}_h^i(t) = \frac{\text{HOD}_i^h(t-1)}{\left(\sum_j \text{FOD}_i^j(t-1) + \sum_h \text{HOD}_i^h(t-1)\right)} * x_i^a(t) \tag{23}$$

Firm $j$ received intermediates to restore its inventories,

$$S_j^{p,\text{restored}}(t) = \sum_{i \to p} \text{FRC}_j^i(t) \tag{24}$$

Therefore, the amount of intermediate $p$ held by firm $i$ at the end of period $t$ is

$$S_j^p(t) = S_j^p(t-1) - S_j^{p,\text{used}}(t) + S_j^{p,\text{restored}} \tag{25}$$

**Demand module.** The demand module represents a characterization of how firms and household issues orders to their suppliers at the end of each period. Firm orders its supplier because of the need to restore its intermediate product inventory. We assume that each firm has a specific target inventory level based on its maximum supply capacity in each time step (see Supplementary Text and Supplementary Table 8 for a sensitivity analysis for different demand structures),

$$S_i^{p,*}(t) = n_i^p * a_i^p * x_i^{\text{max}}(t) \tag{26}$$

Firms issue orders to their suppliers based on their demand and the supply capacity of their suppliers. Then the order issued by firm $i$ to its supplier $j$ is

$$\text{FOD}_j^i(t) = \begin{cases} \left(S_i^{p,*}(t) - S_i^p(t)\right) * \dfrac{\overline{\text{FOD}_j^i} * x_i^a(t)}{\sum_{j \to p}\left(\overline{\text{FOD}_j^i} * x_j^a(t)\right)} & ,\text{if } S_i^{p,*}(t) > S_i^p(t); \\ 0 & \text{if } S_i^{p,*}(t) \le S_i^p(t). \end{cases} \tag{27}$$

Households issue orders to their suppliers based on their demand and the supply capacity of their suppliers. In this study, the demand of household $h$ to final products $q$, $\text{HD}_h^q(t)$, is given exogenously at each time step. Then, the order issued by household $h$ to its supplier $j$ is

$$\text{HOD}_j^h(t) = \text{HD}_h^q(t) * \frac{\overline{\text{HOD}_j^h} * x_j^a(t)}{\sum_{j \to q}\left(\overline{\text{HOD}_j^h} * x_j^a(t)\right)} \tag{28}$$

The total order received by firm $j$ is

$$\text{TOD}_j(t) = \sum_i \text{FOD}_j^i(t) + \sum_h \text{HOD}_j^h(t) \tag{29}$$

**Simulation module.** At each time step, the actions of firms and households are as follows:

1. Firms plan and execute their production based on three factors: (a) inventories of intermediate products they have, (b) supply of primary inputs, and (c) orders from their clients. Firms will maximize their output under these constraints.
2. Product allocation. Firms allocate outputs to clients based on their orders. In equilibrium, the output of firms just meets all orders. When production is constrained by exogenous negative shocks, outputs may not cover all orders. In this case, we use a proportional rationing scheme proposed in the literature[22,23] (see "Allocation module") to allocate products of firms.
3. Firms and household issue orders to their suppliers for the next time step. Firms place orders with their suppliers based on the gaps in their inventories (target inventory level minus existing inventory level). Households place orders with their suppliers

based on their demand. When a product comes from multiple suppliers, the allocation of orders is adjusted according to the production capacity of each supplier.

This discrete-time dynamic procedure can reproduce the equilibrium of the economic system, and can simulate the propagation of exogenous shocks, both from firm and household side, or transportation disruptions, in the economic network. From the firm side, if the supply of a firm's primary inputs is constrained, it will have two effects. On the one hand, the decline in output in this firm means that its clients' orders cannot be fulfilled. This will result in a decrease in inventory of these clients, which will constrain their production. This is the so-called forward or downstream effect. On the other hand, less output in this firm also means less use of intermediate products from its suppliers. This will reduce the number of orders it places on its suppliers, which will further reduce the production level of its suppliers. This is the so-called backward or upstream effect. Similarly, these two effects can also occur if the transport of a firm to its clients or suppliers is restricted. For instance, during the outbreak of COVID-19 in China, the authorities adopted strict isolation measures. These measures have placed constraints on the supply of labor and the transportation of products. This led to a decline in China's output and also triggered the forward and backward effect, which make the shock to propagate to the global economic network. From the household side, the fluctuation of household demand caused by exogenous shocks will also trigger the aforementioned backward effect. Take tourism as an example, during the outbreak of COVID-19 in China, the demand for Chinese tourism from households all over the world will decline significantly. This influence will further propagate to the accommodation and catering industry through supplier-client links.

**Economic footprint.** We define the value-added decrease of all firms in a network caused by an exogenous negative shock as the disaster footprint of the shock. For the firm directly affected by exogenous negative shocks, its loss includes two parts: (a) the value-added decrease caused by exogenous constraints, and (b) the value-added decrease caused by propagation. The former is the direct loss, while the latter is the indirect loss. A negative shock's total economic footprint ($TEF_{i,r}$), direct economic footprint ($DEF_{i,r}$), and propagated economic footprint ($PEF_{i,r}$) for firm $i$ in region $r$ are,

$$TEF_{i,r} = \overline{va}_{i,r} * T - \sum_{t=1}^{T} va_{i,r}^{a}(t) = \overline{va}_{i,r} * T - \sum_{t=1}^{T} x_{i,r}^{a}(t) * b \quad (30)$$

and,

$$DEF_{i,r} = \overline{va}_{i,r} * T - \sum_{t=1}^{T} va_{i,r}^{max}(t) = \overline{va}_{i,r} * T - \sum_{t=1}^{T} x_{i,r}^{max}(t) * b \quad (31)$$

and,

$$PEF_{i,r} = TEF_{i,r} - DEF_{i,r}. \quad (32)$$

We made two simulations and compared the results to obtain the benefits of vaccination. The first simulation is the counterfactual scenario, i.e., a world with no vaccines at all. The results of the first simulation represent global economic loss (includes direct lockdown losses and supply-chain propagations damages) if there is no vaccine. The second simulation is used to calculate the global economic loss under a specific vaccine-distribution scenario. The amount by which the loss of the second simulation is less than the loss of the first simulation is defined as the economic benefit of vaccination.

The lockdown-easing effects for region r, $LEB_r$, is calculated as follows,

$$LEB_r = \sum_i DEF_{i,r}^{Vaccine} - \sum_i DEF_{i,r}^{NoVaccine} \quad (33)$$

and the supply-chain rebuilding benefit for region r, $SRB_r$, is calculated as follows,

$$SRB_r = \sum_i PEF_{i,r}^{Vaccine} - \sum_i PEF_{i,r}^{NoVaccine}. \quad (34)$$

**Global supply-chain network.** We build a global supply-chain network based on version 10 of the Global Trade Analysis Project (GTAP) database[55]. GTAP 10 provides a multiregional input–output (MRIO) table for the year of 2014. This MRIO table divides the world into 141 economies, each of which contains 65 production sectors. If we treat each sector as a firm (producer), and assume that each region has a representative household, we can obtain the following information in the MRIO table: (a) suppliers and clients of each firm; (b) suppliers for each household, and (c) the flow of each supplier-client connection under the equilibrium state. This provides a benchmark for our model.

When applying such a realistic and aggregated network in the disaster footprint model, we need to consider the substitutability of intermediate products supplied by suppliers from the same sector in different regions. The substitution between some intermediate products is fairly straightforward. For example, for a firm that extracts spices from bananas it does not make much of a difference if the bananas are sourced from the Philippines or Thailand. However, for a car manufacturing firm in Japan, which use screw from Chinese auto parts suppliers and engines from German auto parts suppliers to assemble cars, the products of the suppliers in these two regions are non-substitutable. If we assume that all goods are non-substitutable as in the traditional IO model, then we will overestimate the loss of producers such as fragrance extraction firm. If we assume that products from suppliers in the same sector can be completely substitutable, then we will significantly underestimate the losses of producers such as Japanese car manufacturing firm. In order to alleviate the shortcomings of the evaluation deviation under the two assumptions, we set the possibility of substitution for each firm based on the region and sector of supplier supply (see "Allocation module of the model").

## Uncertainty analysis

We performed two uncertainty analysis on vaccine production capacity and model parameters. Supplementary Figs. 12–15 show the distribution of the three types of benefits and the distribution of total benefit, respectively. And Supplementary Figs. 16-19 show the distribution of the three types of benefits and the distribution of total benefit, respectively.

## Data availability

The global dataset used to stimulate the presented results are licensed by the Global Trade Analysis Project at the Center for Global Trade Analysis in Purdue University. The GTAP version 10 can be obtained for a fee from its official website: https://www.gtap.agecon.purdue.edu/databases/v10/index.aspx. Owing to the restriction in the licensing agreement with GTAP, the authors have no right to disclose the original dataset publicly.

## Code availability

The simulation code can be accessed at https://github.com/DaopingW/economic-impact-model. The minimal input for the code is multiregional input–output table. The sample code and test data for the minimal inputs are also provided.

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

## Acknowledgements
This study was supported by the National Natural Science Foundation of China (72091514, 7221101088, 72242105) (D.G.). The authors would like to thank Prashant Yadav and other, anonymous, referee(s) for their helpful comments and suggestions.

## Author contributions
D.W., D.G., and N.C.S. designed the study. O.B., N.C.S., and R.L. developed the epidemiological model. D.W., D.G., and Z.Z. developed the economic model. D.W., R.L., S.Z., and Y.S. performed the analysis. N.C.S., D.G., D.W., T.L., and S.H. interpreted the results. D.W., J.H., and T.L. prepared the figures. D.W., R.L., S.Z., and J.H. prepared the manuscript. D.W., S.Z., and Q.H. prepared the supplementary information. D.G. coordinated, and S.H. and N.C.S. supervised the project.

## Competing interests
The authors declare no competing interests.
