## [Peer Review File · Nature Communications]

Supply chains create global benefits from improved vaccine accessibilityReview Report for Manuscript NCOMMS-22-13893**Title Supply chains create global benefits from improved vaccine accessibility****Summary**

- This paper tries to develop a model to estimate the health gains, gains from lockdown-easing benefit, and benefit from supply-chain flows for different global and national vaccine distribution scenarios. The authors use an epidemiological model and a global trade-model to estimate these benefits.
- Overall, this is an interesting, and extremely policy relevant area of work. I commend the authors for bringing together supply chain/trade, and health pieces together in a meaningful way.
- However, in my opinion the manuscript has included many levels of complexity in one go, which to some degree it appears does not necessarily yield a lot of additional insights. Also, some of the tiered distribution scenarios analyzed and are interlinked but somewhat separate decision problems. Additionally, the paper doesn't carry out detailed/robust sensitivity analysis on vaccine efficacy or disease transmission related parameters. The way population vaccination level impact the production and trade are not illustrated clearly in the manuscript. I put forth some ideas for a revision.

Main Comments

1. The paper is not easy to read/follow given the three tiers of allocation strategies being addressed. Perhaps the authors should first formulate just the global distribution tier without including the additional domestic scenarios based on age and industrial sector. Then show whether balanced distribution and balanced distribution with side transfers from high income non-producing countries can achieve the Pareto outcome. It will make the logic be simpler and flow more clearly. After that add the additional allocation scenarios which take age and industrial sector into account and show how that changes the incentives for balanced distribution and side payment.
2. Much of the stability of vaccine sharing mechanism rests on the high income non vaccine producing countries realizing and internalizing the additional benefits from Balanced distribution. Therefore, it becomes imperative to carry out a robust sensitivity analysis on the benefits by varying key input parameters, especially the vaccine related benefits. The vaccine benefits on health depend on whether the vaccine decreases susceptibility to infection in uninfected people; and/or it decreases severe disease in infected people. Which of these is assumed by the authors is not clear from the epi model on p17 or the parameters in table 12?
3. The authors provide sufficient detail about how they model supply and demand in an interconnected network but there is scant detail on how a given level of vaccination in the population translates into exogenous negative shock? Is it continuous or once a threshold level of vaccination is reached does the shock disappear? Without providing such details it is not possible to fully understand how varying levels of COVID vaccination lead to changes in economic and trade activity through the simulation model used. This may be embedded in the simulation (p25 line 13-15), but it is hard to understand how this is modelled i.e. how does a given level of vaccination change va.
4. The estimates of vaccine production capacity are from sources which doesn't necessarily capture this with high degree or precision. Once again this highlights the need for robust sensitivity analysis on vaccine production capacity across countries. The authors mention that they conducted +25% and -25% sensitivity tests and find that the conclusions in the article will not change. But this is unclear. Did they

vary each country's production capacity by the same amount, did they do a factorial design on the capacity across regions? Such analysis can then also guide decisions on where/which region/country does adding capacity enhance overall global benefits the most?

5. The benefits to vaccine producing countries do not include the manufacturer surplus. It may be small as compared to some of the economic benefits from vaccination and supply chain resumption, but nevertheless has been a point of discussion so worth including that, mentioning that.

Reviewer #2 (Remarks to the Author):

Very well presented paper with clear results, well discussed. I appreciate that this space of modelling does involve many assumptions/parameters/decisions, many of which are addressed appropriately in the discussion of limitations. I do have some concerns relating to the epi part of the model, outlined below, as well as clarity relating to the scenario structure.

Box 1. The presentation of the scenarios should make it clear that they are not independent: really, one scenario involves a decision at scenario set, one of which is global and the other 2 domestic. Defaults are given in the text but should be highlighted somehow in the table. Also, the main text reads "elderly first" and the box reads "old first". I suggest "oldest first" and "youngest first" as a single choice of wording.

Figure 2: use the word "global" when stating the default "Balanced Distribution Strategy", in order to be consistent with box 1. I understand this to be "GIVEN global balance etc., this is what happens under different domestic strategies". But it's not total

The model. In particular, the epi model:

1. If λ and γ are the same for first infection and subsequent infection, I fail to see why you need the S^{np} and I^{np} compartments in the model. Simulations would be no different just putting recovered back in the S class after waning.
2. There is no vaccination in the ODEs, nor does the text explain how Q_i is employed. Really, additional S/I/R classes are required for vaccinated individuals, as vaccine efficacy manifests via different rate parameters through the infection pathway. It is not clear if/how you did this.
3. $a_i=1$ assumes populations stratified by birth year. An age group 0-4, say, that spans 5 years, has ageing rate $1/5$ (if modelled as a Poisson process).
4. Do all rate parameters have units "per day"?
5. Can you justify the choice $R_0=3.5$?
6. Vaccines uniformly distributed over the year - coupled with the statement "the speed of vaccine rollout is equivalent to the easing of the lockdown" (line 27), this seems to imply that lockdown is uniformly eased over the course of 1 year. Can you clarify this please?

Altogether a clear and concise manuscript and a valuable contribution to the literature.

Reviewer #1:

Summary

This paper tries to develop a model to estimate the health gains, gains from lockdown-easing benefit, and benefit from supply-chain flows for different global and national vaccine distribution scenarios. The authors use an epidemiological model and a global trade-model to estimate these benefits.

Overall, this is an interesting, and extremely policy relevant area of work. I commend the authors for bringing together supply chain/trade, and health pieces together in a meaningful way.

However, in my opinion the manuscript has included many levels of complexity in one go, which to some degree it appears does not necessarily yield a lot of additional insights. Also, some of the tiered distribution scenarios analysed and are interlinked but somewhat separate decision problems. Additionally, the paper doesn't carry out detailed/robust sensitivity analysis on vaccine efficacy or disease transmission related parameters. The way population vaccination level impact the production and trade are not illustrated clearly in the manuscript. I put forth some ideas for a revision.

We appreciate the reviewer's positive tone and very constructive comments. By quantifying the multiple trade-off benefits of vaccination, we believe this is the first comprehensively quantitative assessment of the advantage of equitable vaccine distribution. This quantification led to a critical proposal of the benefit-sharing mechanism that can facilitate equitable distributions. Please find our point-by-point response to the comments listed below.

Main Comments

The paper is not easy to read/follow given the three tiers of allocation strategies being addressed. Perhaps the authors should first formulate just the global distribution tier without including the additional domestic scenarios based on age and industrial sector. Then show whether balanced distribution and balanced distribution with side transfers from high income non-producing countries can achieve the Pareto outcome. It will make the logic be simpler and flow more clearly. After that add the additional allocation scenarios which take age and industrial sector into account and show how that changes the incentives for balanced distribution and side payment.

We take the point that the paper is not easy to follow given the three tiers of allocation strategies. Following the reviewer's construction suggestion, we have changed the flow of the results presented. We place the analysis of the results from the Global tier in the first half, including (1) the consequences of different global distribution strategies and (2) the mechanisms that promote the balanced distribution of vaccines globally (P7L8 – P10L14). And then, in the second half, we further (3) analyse how to maximize the effect of limited vaccines under the balanced allocation strategy.

It is worth noting that each scenario is the result of a combination of global and domestic allocation strategies. It is clear that there is no way to calculate the benefits of vaccination if there is only a global allocation strategy and no domestic allocation strategy. Thus, the first half of our results actually still consider a default domestic allocation strategy. Showing only the results under the default domestic allocation strategy in the Global tier analysis makes the manuscript better to be followed. We provide, however, the full results under other domestic allocation strategies in the supporting material (Supplementary Fig. 2-5), to demonstrate the robustness of the analysis of the Global tier.

Much of the stability of vaccine sharing mechanism rests on the high income non vaccine producing countries realizing and internalizing the additional benefits from Balanced distribution. Therefore, it becomes imperative to carry out a robust sensitivity analysis on the benefits by varying key input parameters, especially the vaccine related benefits. The vaccine benefits on health depend on whether the vaccine decreases susceptibility to infection in uninfected people; and/or it decreases severe disease in infected people. Which of these is assumed by the authors is not clear from the epi model on p17 or the parameters in table 12?

Thank you. The uncertainties of the results mainly come from the reliability of the input data (i.e., vaccine production capacities), the parameters of the model, and the setting of key mechanism of the model.

First, for the uncertainty from estimates of vaccine production capacities, we have performed a simple sensitivity analysis in the initial submission by simultaneously scaling up or down the vaccine production capacity of all vaccine-producing countries by 25%. We found that the values in the results are sensitive to vaccine production capacity, but we can draw the same conclusions that 1) the supply chain will amplify the benefits of vaccination and 2) there is a space for Pareto improvement when we consider this cascading effect.

We very much appreciate the reviewer's constructive idea to do sensitivity analysis proposed in his/her second last comment. We take this point and add a Monte Carlo analysis in the revision for uncertainty analysis of vaccine production capacity. We assume that the vaccine production capacity of each vaccine-producing country takes a random value (uniform distribution) within $\pm 25\%$ of its estimated production capacity. And then, we use these random combinations as the initial input to our model. We present the corresponding results and analyse how production capacity will impact the results in SI. Please find it on P33L2 to P35L3 in SI. The graph below shows the distribution of global total benefits. Distribution shifted right from "Selfish Distribution Strategy" to "Balanced Distribution Strategy" scenarios under the default "Oldest" and "High Risk" scenario (left column in Figure 1 below).

Second, for the uncertainty from parameters of the model, we also analysed the sensitivity of the parameters using Monte Carlo approach. Please find the results and corresponding discussion on P35L6 to P37L4 in SI. Distribution shifted right from "Selfish Distribution Strategy" to "Balanced Distribution Strategy" scenarios under the default "Oldest" and "High Risk" scenario (left column in Figure 2 below).

Figure 1 | Distribution of total global benefit under different scenarios

Figure 2 | Distribution of total global benefit under different scenarios

Third, for the uncertainty from the settings of the model, we have already done the sensitivity analysis of the allocation mode of recovery resource and post-disaster demand structure, the two most important sources of uncertainty based on our understanding of the disaster footprint model. Please find the results and corresponding discussion on P38L1 to P38L29 in SI.

For the comment on how the effect of vaccines are introduced into the model, we have added the following description to the method (P18L4 to P19L7; We cited our previous articles in the first submission while did not show full details of the model in the Method):

“For the vaccine intervention case, based on the model shown in Eq(1)-Eq(5), we developed the following realistic age-structured multi-compartmental SEIR model that allows for projections of disease burden of SARS-CoV-2 virus with diverse intervention strategies,

$$\frac{dS_i}{dt} = \omega R_i - \lambda_i S_i - q Q_i S_i \quad (6)$$

$$\frac{dE_i}{dt} = \lambda_i S_i - \delta E_i \quad (7)$$

$$\frac{dI_i}{dt} = \delta E_i - \gamma I_i \quad (8)$$

$$\frac{dR_i}{dt} = \gamma I_i + q Q_i S_i - \omega R_i \quad (9)$$

where S_i , E_i , I_i and R_i are, respectively, the number of susceptible, exposed, infected and recovered individuals in age group i . The recovered/vaccinated individuals are assumed to lose immunity and return to susceptibility after an average protected period of $1/\omega$ and subsequently be liable to reinfection. The average incubation period $1/\delta$ in the analysis is taken to be 5.2 days^{1,2}. To appropriately model the *fraction* of individuals vaccinated, we define the *rate* of vaccination (Q_i) as $-\log(1 - P_i)/D_i$ where P_i and D_i is the vaccine coverage and duration of vaccination in age group i , respectively^{1,3}; q is the vaccine efficacy, take the value 50%.”

The authors provide sufficient detail about how they model supply and demand in an interconnected network but there is scant detail on how a given level of vaccination in the population translates into exogenous negative shock? Is it continuous or once a threshold level of vaccination is reached does the shock disappear? Without providing such details it is not possible to fully understand how varying levels of COVID vaccination lead to changes in economic and trade activity through the simulation model used. This may be embedded in the simulation (p25 line 13-15), but it is hard to understand how this is modelled i.e. how does a given level of vaccination change VA.

Thank you for raising this point. By putting together the assumptions scattered throughout the manuscript, the computation of the lockdown-easing effect can become clearer.

- In the present study, we assume lockdowns will limit the labour supply for a certain percentage (the percentage derived from Google Community Mobility Reports; see Page xx line xx for more details). We assume that this part of the constrained labour supply will ease linearly with vaccine coverage. More intuitively, the people protected by the vaccine can end his isolation and he can start to work. This is a reasonable continuous approximation to discontinuous reality (intermittent lockdown).
- We assume a uniform distribution of vaccine roll-out over the year. This means that the number of people vaccinated also increases linearly.
- And, we assume that when some of the labour supply constraint is lifted, If there happens to be a corresponding demand in the labour market, the VA loss due to this part of labour constraint will not continue.

We have added the following descriptions to the Methods to make the assumptions and calculation clearer.

(In P20 L26 to L30)

“The intensity of a country or region's lockdown at any time period is determined by three factors: its initial lockdown intensity, the size of its population, and the number of people it currently has protected. Among them, the initial lockdown strength is derived from Google Community Mobility Reports (see xx). Population sizes were obtained from the World Bank database. The number of people already protected is calculated from scenarios and our epidemiological model.”

(In P23 L10 to L12)

“For the with-vaccine case, we assume that the constrained labour supply will ease linearly with vaccine coverage. More intuitively, the people protected by the vaccine can end his isolation and he can start to work. Therefore, $\gamma_i^L(t)$ will decrease linearly with vaccine coverage.”

(In P24 L11 to L12)

“Considering that lockdowns only cause labour supply constraint without destroying capital, $(\bar{x}_i - x_i^{max}(t))$ can represents the loss of output due to direct labour supply constraints.”

(In P26 L26 to P27 L2)

We define the value-added decrease of all firms in a network caused by an exogenous negative shock as the disaster footprint of the shock. For the firm directly affected by exogenous negative shocks, its loss includes two parts: a) the value-added decrease caused by exogenous constraints, and b) the value-added decrease caused by propagation. The former is the direct loss, while the latter is the indirect loss. A negative shock's total economic footprint ($TEF_{i,r}$), direct economic footprint ($DEF_{i,r}$), and propagated economic footprint ($PEF_{i,r}$) for firm i in region r are,

$$TEF_{i,r} = \bar{v}a_{i,r} * T - \sum_{t=1}^T va_{i,r}^a(t) = \bar{v}a_{i,r} * T - \sum_{t=1}^T x_{i,r}^a(t) * b$$

and,

$$DEF_{i,r} = \bar{v}a_{i,r} * T - \sum_{t=1}^T va_{i,r}^{max}(t) = \bar{v}a_{i,r} * T - \sum_{t=1}^T x_{i,r}^{max}(t) * b$$

and,

$$PEF_{i,r} = TEF_{i,r} - DEF_{i,r}$$

The estimates of vaccine production capacity are from sources which doesn't necessarily capture this with high degree or precision. Once again this highlights the need for robust sensitivity analysis on vaccine production capacity across countries. The authors mention that they conducted +25% and -25% sensitivity tests and find that the conclusions in the article will not change. But this is unclear. Did they vary each country's production capacity by the same amount, did they do a factorial design on the capacity across regions? Such analysis can then also guide decisions on where/which region/country does adding capacity enhance overall global benefits the most?

We appreciate the creative and constructive comment. As described above, we further analysed the uncertainty from estimates of vaccine production capacity by means of Monte Carlo approach. In the initial submission, we are simultaneously scaling up or down the vaccine production capacity of all vaccine-producing countries by 25%. In the revision, instead, we assume that the vaccine production capacity of each vaccine-producing country takes a random value (uniform distribution) within $\pm 25\%$ of its estimated production capacity. This allows us to capture more possible outcomes.

With regard to the further research about adding capacity in 'where' will enhance overall global benefits the most, we agree that it really a good idea and worth exploring. But it is out of the principal scope of the present study. Therefore, we did not make a more detailed analysis.

The benefits to vaccine producing countries do not include the manufacturer surplus. It may be small as compared to some of the economic benefits from vaccination and supply chain resumption, but nevertheless has been a point of discussion so worth including that, mentioning that.

We thank the reviewer for raising this point and have added some discussion in the manuscript (see P10 L6 to L14). Please see the paragraph below.

"It is worth mentioning that the manufacturer surplus of vaccine-producing countries is not included in the above discussion. Detailed costs for vaccine producers are difficult to obtain. Moreover, some vaccine-producing companies have pledged to provide their doses on a not-for-profit basis until the pandemic ends. And also the annual manufacturer surplus is too small (10 billion order of magnitude) as compared to the health gains and economic benefits from vaccination and supply chain

resumption (1 trillion order of magnitude). If this is taken into account, the benefit-sharing mechanism proposed in the present study will be easier to achieve, as vaccine-producing countries will need less compensation. Hence, when taking the manufacturer surplus into account, all conclusions in the present study will still hold."

Reviewer #2:

Very well-presented paper with clear results, well discussed. I appreciate that this space of modelling does involve many assumptions/parameters/decisions, many of which are addressed appropriately in the discussion of limitations. I do have some concerns relating to the epi part of the model, outlined below, as well as clarity relating to the scenario structure.

We appreciate the constructive comments from the reviewer, which greatly improve our study and make the manuscript clearer.

Box 1. The presentation of the scenarios should make it clear that they are not independent: really, one scenario involves a decision at scenario set, one of which is global and the other 2 domestic. Defaults are given in the text but should be highlighted somehow in the table. Also, the main text reads "elderly first" and the box reads "old first". I suggest "oldest first" and "youngest first" as a single choice of wording.

Thank you. We take the chance to make the manuscript clearer.

- We have added a description of the scenario in box 1
- We highlighted the defaults in box 1
- We revised the scenario names throughout the manuscript to maintain consistency

Figure 2: use the word "global" when stating the default "Balanced Distribution Strategy", in order to be consistent with box 1. I understand this to be "GIVEN global balance etc., this is what happens under different domestic strategies". But it's not total.

We take this point and have revised the manuscript accordingly.

The model. In particular, the epi model:

1. If lambda and gamma are the same for first infection and subsequent infection, I fail to see why you need the S^{np} and I^{np} compartments in the model. Simulations would be no different just putting recovered back in the S class after waning.

Thank you. We made this setting in order to keep the generality of the model, which equipped the model with the ability to handle the case that lambda and gamma are different for first infection and subsequent infection. We explain this setting in Methods as follows (P18 L11 to L15):

“In the case that λ_i and γ are the same for first infection and subsequent infections, we do not need to distinguish between S_i^{np} and I_i^{np} . We set the model as in eq.(1) – eq(5) in order to keep the generality of the model, which equipped the model with the ability to handle the case that λ_i and γ are different for first and subsequent infection”

2. There is no vaccination in the ODEs, nor does the text explain how Q_i is employed. Really, additional S/I/R classes are required for vaccinated individuals, as vaccine efficacy manifests via different rate parameters through the infection pathway. It is not clear if/how you did this.

Thank you for raising this point. Our epidemiological model is based on the study in Li et al (2021a) and Li et al (2021b). We have added the following description to the Methods to clarify how vaccination enters the model (P18 L24 to P19 L7).

“For the vaccine intervention case, based on the model shown in Eq(1)-Eq(5), we developed the following realistic age-structured multi-compartmental SEIR model that allows for projections of disease burden of SARS-CoV-2 virus with diverse intervention strategies,

$$\frac{dS_i}{dt} = \omega R_i - \lambda_i S_i - q Q_i S_i \quad (6)$$

$$\frac{dE_i}{dt} = \lambda_i S_i - \delta E_i \quad (7)$$

$$\frac{dI_i}{dt} = \delta E_i - \gamma I_i \quad (8)$$

$$\frac{dR_i}{dt} = \gamma I_i + q Q_i S_i - \omega R_i \quad (9)$$

where S_i , E_i , I_i and R_i are, respectively, the number of susceptible, exposed, infected and recovered individuals in age group i . The recovered/vaccinated individuals are assumed to lose immunity and return to susceptibility after an average protected period of $1/\omega$ and subsequently be liable to reinfection. The average incubation period $1/\delta$ in the analysis is taken to be 5.2 days^{1,2}. To appropriately model the *fraction* of individuals vaccinated, we define the *rate* of vaccination (Q_i) as $-\log(1 - P_i)/D_i$ where P_i and D_i is the vaccine coverage and duration of vaccination in age group i , respectively^{1,3}; q is the vaccine efficacy, take the value 50%.”

3. $a_i=1$ assumes populations stratified by birth year. An age group 0-4, say, that span 5 years, has ageing rate 1/5 (if modelled as a Poisson process).

Yes, it is. For example, in Eq(1),

$$\frac{dS_i^p}{dt} = a_{i-1} S_{i-1}^p - \lambda_i S_i^p - a_i S_i^p \quad (1)$$

$a_{i-1}S_{i-1}^p$ represent 1/5 of S_{i-1}^p will aging in group i , while $a_i S_i^p$ represent 1/5 of S_i^p will aging out group i .

4. Do all rate parameters have units "per day"?

Yes. All rate parameters have units "per day". We appreciate the comment and make this clear in our paper now.

5. Can you justify the choice $R_0=3.5$?

We made our choice based on the R_0 reported by some of the existing literature. For example, Liu et al identified 12 studies which estimated the basic reproductive number for COVID-19 from China and overseas and shown that the estimates ranged from 1.4 to 6.49. We have added corresponding explanations to the article (P18 L19 to L21).

Liu, Ying, Albert A. Gayle, Annelies Wilder-Smith, and Joacim Rocklöv. "The reproductive number of COVID-19 is higher compared to SARS coronavirus." Journal of travel medicine (2020).

"For R_0 , we refer to Liu et al, in which identified 12 studies which estimated the basic reproductive number for COVID-19 from China and overseas and shown that the estimates ranged from 1.4 to 6.49."

6. Vaccines uniformly distributed over the year - coupled with the statement "the speed of vaccine rollout is equivalent to the easing of the lockdown" (line 27), this seems to imply that lockdown is uniformly eased over the course of 1 year. Can you clarify this please?

Thank you for raising this point. We assume that lockdowns gradually eased with vaccination. This is a reasonable continuous approximation to discontinuous reality. But it's worth noting that the lockdown wasn't completely lifted within 1 year. The intensity of a country or region's lockdown at any time period is determined by three factors:

- its initial lockdown intensity,
- the size of its population,
- and the number of people it currently has protected.

Among them, the initial lockdown strength is derived from Google Community Mobility Reports (see xx). Population sizes were obtained from the World Bank database. The number of people already protected is calculated from scenarios and our epidemiological model.

We have added some description in the Method (P20 L26 to L30). Please see the paragraph below.

"The intensity of a country or region's lockdown at any time period is determined by three factors: its initial lockdown intensity, the size of its population, and the number of people it currently has protected. Among them, the initial lockdown strength is derived from Google Community Mobility Reports (see xx). Population sizes were obtained from the World Bank database. The number of people already protected is calculated from scenarios and our epidemiological model."

Altogether a clear and concise manuscript and a valuable contribution to the literature.

Again, we very much appreciate the comments we received, which are very useful and greatly improve our paper.

- 1 Li, R., Bjørnstad, O. N. & Stenseth, N. C. Switching vaccination among target groups to achieve improved long-lasting benefits. *Royal Society open science* **8**, 210292 (2021).
- 2 Li, R., Metcalf, C. J. E., Stenseth, N. C. & Bjørnstad, O. N. A general model for the demographic signatures of the transition from pandemic emergence to endemicity. *Science Advances* **7**, eabf9040, doi:10.1126/sciadv.abf9040 (2021).
- 3 Bjørnstad, O. N. *Epidemics: models and data using R*. (Springer, 2018).

REVIEWERS' COMMENTS

Reviewer #1 (Remarks to the Author):

The authors have successfully addressed most of my comments on the earlier version of this manuscript. Where it was not feasible, they have provided adequate responses.

My only remaining comment is:

Where/which regional locations will adding more production capacity contribute the most to the objective function of the global allocation--is an important policy question. I understand the authors cannot address this with the sensitivity analysis carried out in this paper. It is important to acknowledge this as an area for future research or point to any other existing opinion papers or articles which may point to this.

Thanks,
Prashant Yadav

Reviewer #2 (Remarks to the Author):

I am happy that my comments have been addressed.

Reviewer #1 (Remarks to the Author):

The authors have successfully addressed most of my comments on the earlier version of this manuscript. Where it was not feasible, they have provided adequate responses.

My only remaining comment is:

Where/which regional locations will be adding more production capacity contribute the most to the objective function of the global allocation--is an important policy question. I understand the authors cannot address this with the sensitivity analysis carried out in this paper. It is important to acknowledge this as an area for future research or point to any other existing opinion papers or articles which may point to this.

Thanks,

Prashant Yadav

Thank you very much for the constructive comments. We do agree that "Where/which regional locations will be adding more production capacity contribute the most to the objective function of the global allocation" is an important policy question. We have pointed out this as an area for future research in the limitation part of the manuscript.

"In addition, exploring where/which regional locations will be adding more production capacity contribute the most to the objective function of the global allocation is also an important and interest policy question but not include in the present study. This is an area worth exploring in the future."

Reviewer #2 (Remarks to the Author):

I am happy that my comments have been addressed.

We appreciate your constructive comments during the review process. Many thank.